# UNIFIED NEURAL SCALING LAWS

## ABSTRACT

We present a functional form (that we refer to as a *Unified Neural Scaling Law (UNSL)*) that accurately models and extrapolates the scaling behaviors of deep neural networks as multiple dimensions all vary simultaneously (i.e. how the evaluation metric of interest varies as one simultaneously varies the number of model parameters, training dataset size, number of training steps, and various hyperparameters) for various architectures and for each of various tasks within a varied set of upstream and downstream tasks. When compared to other functional forms for neural scaling, this functional form yields extrapolations of scaling behavior that are considerably more accurate on this set.

## 1 INTRODUCTION

Training today's state-of-the-art neural networks requires significant amounts of computational resources and training data. Given a wide range of available methods and architectures to choose from, accurate forecasting of their performance is essential for selecting those that are likely to perform best at scale, especially since the top-performing methods at smaller scales often fail to maintain their performance at larger scales (Sutton, 2019; Tolstikhin et al., 2021). Moreover, accurate forecasting of neural network behaviors at scale is critical not only for identifying the top-performing approaches but also for ensuring AI safety, as predicting the emergence of novel capabilities at scale is essential for responsible development and deployment of advanced AI systems. This realization motivated the study of *neural scaling laws* (Cortes et al., 1994; Hestness et al., 2017; Rosenfeld et al., 2019; Kaplan et al., 2020; Zhai et al., 2021; Abnar et al., 2021; Brown et al., 2020; Bahri et al., 2021; Alabdulmohsin et al., 2022; Caballero et al., 2023) which aim to predict the behavior of large-scale models as the amount of compute, data, and model parameters increases.

Clearly, the accuracy, as well as the confidence of predictions made by neural scaling laws can only increase (or remain the same) as a larger number of relevant predictors are included, due to the standard conditional entropy inequality, $H(Y|\mathbf{X}) \leq H(Y)$, where $\mathbf{X}$ is the vector of predictive variables and $Y$ is the performance evaluation metric. Namely, as the number of predictive variables $X_i, i = 1, ..., m$ increases, the conditional entropy $H(Y|(X_1, ..., X_m))$ can only decrease (or remain the same). Ultimately, to obtain the maximal achievable reduction in the entropy of $Y$, one would need to identify the set of all possible $X_i$ that are causally related to $Y$, and develop a complete model $P(Y|\mathbf{X})$ that can serve as a "unified functional form" of neural network behavior(s) at scale.

To address this need for a (more) unified functional form, we present *Unified Neural Scaling Laws (UNSL)*, a functional form that accurately models and extrapolates the scaling behaviors of deep neural networks as multiple dimensions all vary simultaneously. When compared to other functional forms for neural scaling, this functional form yields extrapolations of scaling behavior that are considerably more accurate on this set. Moreover, this functional form accurately models and extrapolates multivariate scaling behavior that other functional forms are incapable of expressing such as the nonmonotonic transitions present in the scaling behavior of overfitting and hyperparameters (such as learning rate and standard deviation of weights at initialization) that have a nonmonotonic relationship with the performance evaluation metric.

## 2 THE FUNCTIONAL FORM OF UNIFIED NEURAL SCALING LAWS

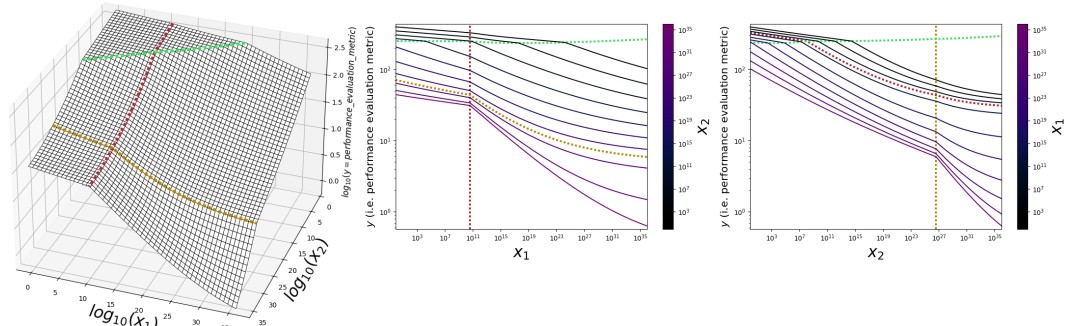

Figure 1: An illustration of a Unified Neural Scaling Law (UNSL) (dark solid lines) with two input dimensions, $x_1$ and $x_2$; the central and the right plots show the projections on each of the input dimensions, respectively. In this particular example, an UNSL contains 3 hyperbreaks highlighted by brighter dotted lines - orange, yellow, and green. The green hyperbreak is created by a non-bottleneck component. The orange hyperbreak is created by an $x_1$ bottleneck component. The yellow hyperbreak is created by an $x_2$ bottleneck component. See Section 2 for detailed explanation of hyperbreaks.

Let $y$ denote a performance evaluation metric of interest, e.g. prediction error or cross-entropy, "upstream" (i.e., measured on the validation dataset from the pretraining data distribution) or "downstream" (i.e., measured on new data and/or tasks that the model does not encounter during pretraining), and let $(x_i)_{i=1}^m \in \overline{\mathbb{R}}_{>0}^m$ denote a tuple of $m$ quantities that can be viewed as predictors of $y$, e.g. number of model parameters, training dataset size, number of training steps, and values of various hyperparameters.

We present the following general functional form of a unified neural scaling law (UNSL):

$$y = (a_0 + Q(2)) + \underbrace{\left(Q(S+3) + a_1^{-1}\right)^{-1}}_{\text{oppositional force of overfitting}},$$ (1)

where $Q$ is defined as follows:

$$Q(q) = \left((R(q))^{-1} + a_q^{-1}\right)^{-1} + \underbrace{\sum_{s=1}^{S}\left(R(q+s) + a_{q+s}^{-1}\right)^{-1}}_{\text{oppositional force of hyperparameters}},$$ (2)

where $R$ is defined as follows:

$$R(r) = \underbrace{K(U_r,\ n_{r_0},\ r\cdot(m+1))}_{\text{non-bottleneck component}} + \underbrace{\sum_{t\in T_r} K(\{t\},\ n_{r_t},\ r\cdot(m+1)+t)}_{\text{bottleneck components}},$$ (3)

where $U_r, T_r \subseteq \{1,\dots,m\}$,

and where $K$ is a *Multivariate Broken Neural Scaling Law (MBNSL)*, defined as follows:

$$K(M,n,k) = b_k \cdot \left(\prod_{i\in M} x_i^{-c_{i_{0_k}}}\right) \prod_{j=1}^{n}\left(1 + \left(\frac{\prod_{i\in M} x_i^{c_{i_{j_k}}}}{d_{j_k}}\right)^{\left|\frac{1}{f_{j_k}}\right|}\right)^{-f_{j_k}}.$$ (4)

The parameters whose values are unknown constants that must be estimated by fitting the above functional form to the $(x_1 \dots x_m, y)$ data points are all those whose base is one of these: $a, b, c, d, f$. The purpose of the variables $i, j, k, q, r, s, t$ is indexation. $n$ is a bound of a product operator; as a result, each of $n_{r_0}$ and $n_{r_t}$ implicitly is a bound of a product operator. $S$ is a bound of a summation operator. $M \subseteq \{1,\dots,m\}$. $M$ is a product index set; as a result, $U_r$ implicitly is a product index

set. $T_r$ is a summation index set. $K, Q, R$ are functions and the contents of the parentheses in $K(\cdot), Q(\cdot), R(\cdot)$ are arguments of those functions. Whenever an argument of $K$, $Q$, or $R$ is obtained via addition(s) and/or multiplication(s), the sole reason that those additions and multiplications occur is to cause each instantiation of $K$ to have a unique value for $k$.

**Equations 1, 2, 3, and 4 are interpreted as follows.**

We use the term ***multi-log space*** to refer to the (m+1)-dimensional space obtained by applying the logarithmic transformation to each of every dimension $(x_1...x_m, y)$.

**Equation 4** is an extension of the univariate *broken neural scaling law (BNSL)* of Caballero et al. (2023) to multivariate settings. When $|M| = 1$, its expressivity is identical to the univariate broken neural scaling law functional form (with the performance limit term subtracted out) from Caballero et al. (2023). When $|M| > 1$, Equation 4 defines a sequence of $n + 1$ smoothly connected hyperplanes in multi-log space. Constant $n$ corresponds to the number of (smooth) "hyperbreaks" (i.e. transitions) between $n + 1$ consecutive hyperplanes in multi-log space; the dimensionality of each hyperplane is $|M|$, and the dimensionality of each hyperbreak is $|M| - 1$. When $n = 0$, Equation 4 becomes $b_k \prod_{i \in M} x_i^{-c_{i_{0_k}}}$. In multi-log space, the initial exponent for each input dimension $(c_{i_{0_k}})_{i \in M}$ corresponds to the gradient of the first hyperplane with respect to the input dimensions $(x_i)_{i \in M}$. In multi-log space, $b_k$ corresponds to the offset of the output of Equation 4. The j-th hyperplane smoothly transitions to the (j+1)th hyperplane at the values of $(x_i)_{i \in M}$ for which this equality is true: $d_{j_k} = \prod_{i \in M} x_i^{c_{i_{j_k}}}$. The j-th exponent for each input dimension $(c_{i_{j_k}})_{i \in M}$ multiplied by $\text{sign}(f_{j_k})$ corresponds to the change in gradient (with respect to the input dimensions $(x_i)_{i \in M}$) between the j-th hyperplane and the (j+1)th hyperplane in multi-log space. Constant $f_{j_k}$ represents the sharpness of the hyperbreak between the j-th and the (j+1)th hyperplane in multi-log space; smaller values of $|f_{j_k}|$ yield a sharper hyperbreak and regions (before and after the j-th hyperbreak) that have less curvature in multi-log space; larger values of $|f_{j_k}|$ yield a smoother (wider) hyperbreak and regions (before and after the j-th hyperbreak) that have more curvature in multi-log space.

**Equation 3** consists of 2 kinds of components. The component $K(U_r,\ n_{r_0},\ r \cdot (m+1))$ is referred to as a "non-bottleneck" component and corresponds to the smoothly connected hyperplanes (in multi-log space) as described in the previous paragraph. Each of the components summed together in the summation $\sum_{t \in T_r} K(\{t\},\ n_{r_t},\ r \cdot (m+1) + t)$ is referred to as a "bottleneck" component and corresponds to each of the performance limits when bottlenecked by each of the dimensions $(x_t)_{t \in T_r}$.

**Equation 2** is as follows. $R(q)$ represents everything that has been discussed thus far in this Section 2; $a_q$ represents a misperformance limit (e.g., the cross-entropy or test error rate of random guessing). The remaining contents of Equation 2 represent the "oppositional force" of hyperparameters (such as learning rate and standard deviation of weights at initialization) that have an oppositional relationship with the performance evaluation metric; for example, when learning rate and/or standard deviation of weights at initialization are too large, they exert an "oppositional force" on the value of $Q(q)$. $S$ represents the number of misperformance limits of the "oppositional force" of hyperparameters.

**Equation 1** consists of two pairs of parentheses. The contents of the left pair of parentheses represent everything that has been discussed thus far in this Section 2, plus the constant $a_0$; $a_0$ corresponds to the limit as to how far the value of $y$ can be reduced (or maximized) even if all of $x_1...x_m$ go to the values of $(x_i)_{i=1}^m \in \overline{\mathbb{R}}_{>0}^m$ that yield the global optimum of $y$. The contents of the right pair of parentheses are identical to the contents of the left pair except for the fact that all of the constants are completely different than the constants in the first set of parentheses. Additionally, the right pair of parentheses is raised to the $-1$ power. The purpose of the contents of the right pair of parentheses is to represent the "oppositional force" exerted by overfitting; for example, when one trains a model for more than one epoch, the contents of this pair of parentheses (raised to the $-1$ power) become a non-negligible number that is considerably larger than zero.

## 2.1 THE ADDITIVE SYMMETRY

The following expression[1] implicitly shows up in several places (when an addition takes place) in Equations 1, 2, and 3:

---

[1] In Equation 5, $b$, $c_{i_0}$, $g$, and $h_i$ are constants estimated by fitting Equation 5 to $(x_1...x_m, y)$ data points.

$$y = b \cdot \left( \prod_{i=1}^{m} x_i^{-c_{i_0}} \right) + g \cdot \left( \prod_{i=1}^{m} x_i^{h_i} \right), \tag{5}$$

and is equivalent to a ($n = 1, M = \{1, \ldots, m\}$) version of Equation 4:

$$y = b \cdot \left( \prod_{i=1}^{m} x_i^{-c_{i_0}} \right) \left( 1 + \left( \frac{\prod_{i=1}^{m} x_i^{c_{i_1}}}{d} \right)^{\left| \frac{1}{f} \right|} \right)^{-f}, \tag{6}$$

when all these equalities are simultaneously true:

$$f = -1 \quad , \qquad c_{i_1} = c_{i_0} + h_i \quad , \qquad d = b/g \quad .$$

Equation 5 is different from Equation 6 in that (assuming $b$, $g$, and $d$ are positive numbers):

1. For Equation 5, the change in gradient (with respect to the input dimensions $x_1, \ldots, x_m$ as any $x_i$ increases) between the 1st hyperplane and the 2nd hyperplane in multi-log space is always nonnegative; meanwhile, for Equation 6, this change in gradient can be any amount.

2. For Equation 5, the sharpness of the hyperbreak between the 1st and the 2nd hyperplane in multi-log space is dependent solely on the amount of change in gradient between the 1st hyperplane and the 2nd hyperplane in multi-log space; meanwhile, for Equation 6, this sharpness is dependent on the value of $f$ (and as a result is decoupled from the amount of change in gradient between the 1st hyperplane and the 2nd hyperplane in multi-log space).

Empirically, we observe that nonmonotonic transitions always seem to be characterized by Equation 5 rather than 6. As a result, (when an addition takes place in the center) in Equations 1 and 2, we implicitly use Equation 5 to model phenomena (e.g. overfitting and hyperparameters such as learning rate and standard deviation of weights at initialization) that are capable of exhibiting a nonmonotonic relationship with the performance evaluation metric.

Empirically, we observe that transitions to or from regions in which the gradient (with respect to at least one of the input dimensions $x_1, \ldots, x_m$) is equal to zero always seem to be characterized by a version of Equation 5 in which each $h_i$ (in $h_1, \ldots, h_m$) for which the gradient with respect to $x_i$ (in $x_1, \ldots, x_m$) is equal to zero is equal to zero. As a result, we implicitly use that version of Equation 5 when addition takes place in Equation 3 and when addition takes place with a parameter whose base is $a$ in parts of Equations 1 and 2.

Note that Equation 5 sums two $n = 0$ versions of MBNSL of Equation 4. To extend the relations discussed in this Section 2.1 thus far to a summation of two MBNSLs that each have an arbitrary number of hyperbreaks $n$, see Appendix 7.

## 2.2 DESIDERATA

The UNSL functional form satisfies all of the following desiderata:

1. Each univariate scaling behavior is a univariate *broken neural scaling law (BNSL)* of Caballero et al. (2023). This means that (as discussed in Section 2.1) for a significant subset of transitions between consecutive hyperplanes (in multi-log space) the sharpness needs to be decoupled from the amount of change in gradient (i.e. via the extra expressivity granted by $f$ in Equation 6 (and Equation 4)).

2. The position of break(s) (within univariate scaling behaviors) within hyperbreak(s) created by non-bottleneck components are shifted via multiplication in a way that is dependent on other input dimensions.

3. Whenever all but one $x_i$ dimension in $x_1 \ldots x_m$ simultaneously go to the values of $(x_i)_{i=1}^{m} \in \overline{\mathbb{R}}_{>0}^{m}$ that yield the global optimum of $y$, that performance limit is dependent on the value of that single $x_i$ dimension (that is bottlenecking performance) and no other dimension in $x_1 \ldots x_m$. When sufficiently close to the global optimum of $y$, the transition to that performance limit is characterized by the functional form $y = a + \sum_{t \in T_r} b_t \cdot x_t^{-c_t}$.

4. The performance limit as all $x_i$ dimensions in $x_1...x_m$ simultaneously go to the values of $(x_i)_{i=1}^m \in \overline{\mathbb{R}}_{>0}^m$ that yield the global optimum of $y$ is dependent on a constant (e.g. the irreducible entropy or bayes error). The transition to this performance limit is characterized by summing an entire functional form with a constant (e.g. $a_0$).

5. The misperformance limit (e.g. upper limits when using metrics such as error or loss for which a lower value of that metric is considered better) when the amount of misperformance is not bottlenecked by any $x_i$ in $x_1...x_m$ is dependent on a constant. The transition to this misperformance limit is characterized by raising to the -1 power the sum of a functional form and a constant. Examples of such misperformance limits in some scenarios are the loss or error of a random guessing (maximum entropy) model and in other scenarios are a value much larger than the loss or error of a random guessing (maximum entropy) model.

6. Whenever all but one $x_i$ dimension in $x_1...x_m$ simultaneously go to the values of $(x_i)_{i=1}^m \in \overline{\mathbb{R}}_{>0}^m$ that yield the globally worst value of $y$, that misperformance limit (e.g. upper limits when using metrics such as error or loss for which a lower value of that metric is considered better) is dependent on the value of that single $x_i$ dimension (that is bottlenecking misperformance) and no other dimension in $x_1...x_m$. When sufficiently far from the global optimum of $y$, the transition to that misperformance limit is characterized by the functional form $y = \left(a + \sum_{t \in T_r} b_t \cdot x_t^{-c_t}\right)^{-1}$. Examples of such misperformance limits are the high loss or error obtained when training dataset size is too small (i.e. such that overfitting occurs).

7. Nonmonotonic transitions (e.g. due to overfitting and hyperparameters such as learning rate and standard deviation of weights at initialization) are characterized by the additive symmetry discussed in Section 2.1.

8. The "oppositional forces" of hyperparameters oppose "good learning" (i.e. the subset of learning that is not considered to be overfitting) and "bad learning" (i.e. the subset of learning that is considered to be overfitting).

**See Appendix 13 for explanation of how UNSL functional form satisfies all of these desiderata. See Appendix 14 for evidence that all of these desiderata are empirically true.**

## 3  RELATED WORK

To the best of our knowledge, Rosenfeld et al. (2019) was the first to describe a functional form with multivariate input; this functional form is $y = a + b_1 x_1^{-c_1} + b_2 x_2^{-c_2}$ in which $x_1$ is number of model parameters and $x_2$ is training dataset size. Kaplan et al. (2020) (and others such as Hoffmann et al. (2022)) used this same functional form, but had $x_2$ be number of training steps multiplied by training batch size; we refer to this functional form as "CF".

Muennighoff et al. (2023) introduced this functional form (that we refer to as "DC") with trivariate input:

$$y = a + b_1 \cdot (U_N + U_N \cdot d_1 \cdot (1 - e^{(-1 \cdot R_N/(d_1))}))^{-c_1} + b_2 \cdot (x_3 + x_3 \cdot d_2 \cdot (1 - e^{(-1 \cdot R_D/(d_2))}))^{-c_2} ;$$

in that functional form:

$$R_D = \max(0, (x_2/x_3) - 1),$$

$$U_N = \min(x_1, (x_3 \cdot ((c_1 \cdot b_1)/(c_2 \cdot b_2))^{(1/(c_1+c_2))})^{(c_2/c_1)} \cdot ((c_1 \cdot b_1)/(c_2 \cdot b_2))^{(1/(c_1+c_2))}),$$

$$R_N = \max(0, (x_1/U_N) - 1),$$

$x_1$ is number of model parameters, $x_2$ is number of training steps multiplied by training batch size, and $x_3$ is training dataset size. When training dataset size is so large that one only trains for one epoch, functional form "DC" is mathematically identical to functional form "CF".

# 4 EMPIRICAL RESULTS: FITS & EXTRAPOLATIONS OF FUNCTIONAL FORMS

We now show the fits & **extrapolations** of various functional forms. **In all plots here, onward, & in the appendix, triangle-shaped points are points used for fitting a functional form, circle-shaped points are held-out points used for evaluating extrapolation of functional form fit to the triangle-shaped points, & lines are the functional form that has been fit to triangle-shaped points. The color of each line and (the inside of) each point represents its value along the color bar dimension. Lines of the functional form are intentionally only rendered at the values of the color bar dimension for which there exists at least one (triangle-shaped or circle-shaped) point; this means that the vertical distance of each point from the line (that is the same color as that point) represents the error of the functional form when fitting (or extrapolating to) that point. 100% of the plots in this paper here, onward, & in the appendix contain circle-shaped point(s) for evaluating extrapolation.**

See Section 10 for details on fitting UNSL. See Appendix 12 for an analysis of how the number of observed points used for fitting affects extrapolation accuracy.

All the extrapolation evaluations reported in the tables (that have ↓ symbol in the top row) are reported in terms of root mean squared log error (RMSLE) ± root standard log error. See Appendix 8 for definition of RMSLE and Appendix 9 for definition of root standard log error.

### 4.0.1 ABLATION FUNCTIONAL FORMS

A1 functional form refers to the baseline ablation functional form in which all the additive symmetries discussed in Section 2.1 have been removed such that this A1 baseline functional form is Equation 4, i.e. :

$$y = b \cdot \left( \prod_{i=1}^{m} x_i^{-c_{i_0}} \right) \prod_{j=1}^{n} \left( 1 + \left( \frac{\prod_{i=1}^{m} x_i^{c_{i_j}}}{d_j} \right)^{\left| \frac{1}{f_j} \right|} \right)^{-f_j} .$$

A2 functional form refers to the baseline ablation functional form that consists solely of Equation 3 (which consists of Equation 4) plus the constant $a_0$ (which corresponds to the limit as to how far the value of $y$ can be reduced (or maximized) even if all of $x_1...x_m$ go to the values of $(x_i)_{i=1}^{m} \in \overline{\mathbb{R}}_{>0}^{m}$ that yield the global optimum of $y$):

$$y = a_0 + K(U_0, \ n_{0_0}, \ 0) + \sum_{t \in T_0} K(\{t\}, \ n_{0_t}, \ t), \quad \text{where } U_0, T_0 \subseteq \{1, \ldots, m\}.$$

A2 functional form incorporates more of the additive symmetries discussed in Section 2.1 than A1 functional form does.

A3 functional form refers to the baseline ablation functional form that consists solely of Equation 2 (which consists of Equation 3 (which consists of Equation 4)) plus the constant $a_0$ (which corresponds to the limit as to how far the value of $y$ can be reduced (or maximized) even if all of $x_1...x_m$ go to the values of $(x_i)_{i=1}^{m} \in \overline{\mathbb{R}}_{>0}^{m}$ that yield the global optimum of $y$):

$$y = a_0 + \left( (R(0))^{-1} + a_1^{-1} \right)^{-1} + \sum_{s=1}^{S} \left( R(s) + a_{s+1}^{-1} \right)^{-1} .$$

A3 functional form incorporates more of the additive symmetries discussed in Section 2.1 than A2 functional form does. UNSL functional form incorporates all of the additive symmetries discussed in Section 2.1 (i.e. more than A3 functional form does).

### 4.0.2 SUMMARY OF RESULTS

| Domain | CF ↑ | DC ↑ | A1 ↑ | A2 ↑ | A3 ↑ | UNSL ↑ |
|---|---|---|---|---|---|---|
| Downstream Image Classification | 0.00% | 0.00% | 8.70% | 8.70% | 21.74% | **60.87%** |
| Language (Downstream & Upstream) | 0.00% | 0.00% | 0.00% | 11.11% | 0.00% | **88.89%** |

Table 1: Percentage of tasks by domain where each functional form is the best for **extrapolation** of scaling behavior. See Sections 4.1 and 4.2 for more details.

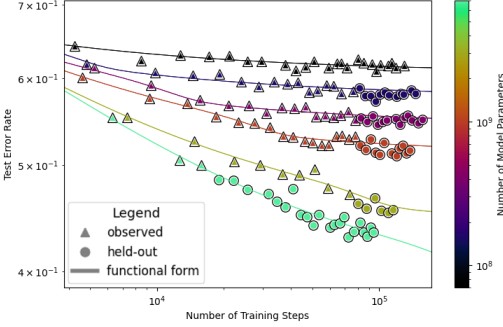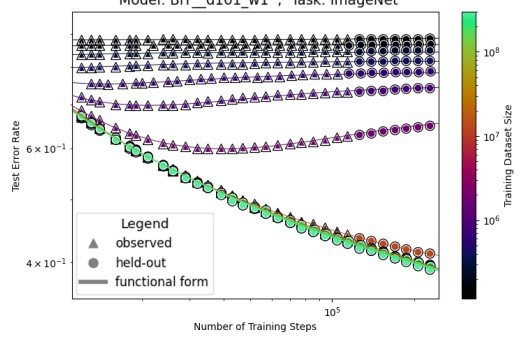

Figure 2: UNSL accurately Extrapolating Downstream Performance; there are many additional accurate extrapolation results in Appendix 15. Experimental data of scaling behavior in left plot is downstream performance on CSR (Common Sense Reasoning), i.e. downstream zero-shot mean test error rate across HellaSwag, ARC (easy and challenge), PIQA, WinoGrande, OpenBookQA, SIQA, and BoolQ; see Section 4.2 for more details. Experimental data of scaling behavior in right plot is few-shot downstream performance on ImageNet; see Section 4.1 for more details.

### 4.1 VISION

We evaluate how well various functional forms extrapolate performance on downstream vision tasks as multiple dimensions vary simultaneously. The tasks that are evaluated are test error rate on each of various few-shot downstream image classification tasks; the downstream tasks are: Birds 200 (Welinder et al., 2010), Cars 196 (Krause et al., 2013), and ImageNet (Deng et al., 2009). The following architectures of various sizes are pre-trained on subsets of JFT-300M (Sun et al., 2017): vision transformers (ViT) (Dosovitskiy et al., 2020), MLP mixers (MiX) (Tolstikhin et al., 2021), and big-transfer residual neural networks (BiT) (Kolesnikov et al., 2020). The bivariate subset of this scaling behavior data is obtained via correspondence with authors of Alabdulmohsin et al. (2022); the simultaneously varying dimensions of the bivariate scaling behavior are training dataset size and number of training steps. The trivariate subset of this scaling behavior data is obtained from the ViT/16 results of Zhai et al. (2022); the simultaneously varying dimensions of the trivariate scaling behavior are training dataset size, number of training steps, and number of model parameters. As can be seen in Tables 1, 2, and 3, UNSL yields extrapolations with the lowest RMSLE (Root Mean Squared Logarithmic Error) for 60.87% of tasks of any of the functional forms, while the next best functional form performs the best on only 21.74% of the tasks. To view plots of UNSL, DC, A1, A2, and A3 on each of these bivariate scaling behaviors, in Appendix 15.1 respectively see Figures 11, 12, 13, 14, 15. To view plots of UNSL, DC, A1, A2, and A3 on each of these trivariate scaling behaviors, in Appendix 15.1 respectively see Figures 16, 17, 18, 19, 20.

### 4.2 LANGUAGE

We evaluate how well various functional forms extrapolate performance on downstream (and up-stream) language tasks as multiple dimensions vary simultaneously. As can be seen in Tables 1, 5, and 4, UNSL yields extrapolations with the lowest RMSLE (Root Mean Squared Logarithmic Error) for 88.89% of tasks of any of the functional forms, while the next best functional form performs the best on only 11.11% of the tasks. To view plots of UNSL, DC, A1, A2, and A3 on trivariate scaling

| Task | Model | DC ↓ | A1 ↓ | A2 ↓ | A3 ↓ | UNSL ↓ |
|------|-------|------|------|------|------|--------|
| Birds | BiT/d101/w3 | 3.97e-1 ± 9.84e-3 | 1.80e-2 ± 1.16e-3 | 2.12e-2 ± 1.43e-3 | 1.60e-2 ± 1.34e-3 | **1.28e-2 ± 9.05e-4** |
| Birds | BiT/d50/w3 | 4.45e-1 ± 1.16e-2 | 4.29e-2 ± 2.94e-3 | 1.67e-2 ± 1.13e-3 | **1.54e-2 ± 1.15e-3** | 1.63e-2 ± 1.48e-3 |
| Birds | MiX/L/16 | 4.92e-1 ± 1.15e-2 | **1.39e-2 ± 8.46e-4** | 1.72e-2 ± 1.05e-3 | 9.85e-2 ± 7.86e-3 | 1.47e-2 ± 9.82e-4 |
| Birds | MiX/B/16 | 2.12e-1 ± 1.88e-3 | 2.05e-2 ± 1.44e-3 | 2.02e-2 ± 2.12e-3 | **1.75e-2 ± 1.29e-3** |
| Birds | BiT/d50/w1 | 3.50e-1 ± 9.39e-3 | 1.16e-2 ± 7.67e-4 | 1.36e-2 ± 1.03e-3 | **1.15e-2 ± 7.48e-4** | 1.56e-2 ± 1.29e-3 |
| Birds | ViT/B/16 | 3.40e-1 ± 8.03e-3 | 5.86e-2 ± 6.60e-3 | 3.08e-2 ± 1.57e-3 | 1.64e-2 ± 8.98e-4 | **1.58e-2 ± 1.04e-3** |
| Birds | BiT/d101/w1 | 3.97e-1 ± 9.84e-3 | 1.80e-2 ± 1.16e-3 | 1.29e-2 ± 9.28e-4 | **1.23e-2 ± 9.23e-4** | 1.28e-2 ± 9.05e-4 |
| Cars | MiX/L/16 | 6.23e-1 ± 1.36e-2 | 5.83e-2 ± 5.45e-3 | 4.54e-2 ± 2.49e-3 | 3.50e-2 ± 3.01e-3 | **2.73e-2 ± 2.22e-3** |
| Cars | MiX/B/16 | 7.05e-1 ± 1.42e-2 | 3.96e-2 ± 2.42e-3 | 2.46e-2 ± 2.15e-3 | 3.27e-2 ± 3.28e-3 | **2.32e-2 ± 1.87e-3** |
| Cars | ViT/B/16 | 1.05e+0 ± 1.64e-2 | 1.36e-1 ± 9.15e-3 | 8.74e-2 ± 4.71e-3 | 9.76e-2 ± 7.18e-3 | **2.67e-2 ± 1.69e-3** |
| Cars | BiT/d101/w3 | 3.03e-1 ± 7.80e-3 | 2.24e-2 ± 1.61e-3 | 2.12e-2 ± 1.43e-3 | 1.80e-2 ± 1.44e-3 | **1.75e-2 ± 1.34e-3** |
| Cars | BiT/d101/w1 | 5.91e-1 ± 1.02e-2 | 3.89e-2 ± 1.97e-3 | 2.77e-2 ± 1.68e-3 | **2.29e-2 ± 1.86e-3** | 2.90e-2 ± 1.75e-3 |
| Cars | BiT/d50/w3 | 3.87e-1 ± 1.29e-2 | 2.66e-2 ± 2.05e-3 | **2.55e-2 ± 2.00e-3** | 2.75e-2 ± 2.23e-3 | 2.62e-2 ± 2.10e-3 |
| Cars | BiT/d50/w1 | 6.71e-1 ± 1.32e-2 | 1.99e-2 ± 1.45e-3 | **1.93e-2 ± 1.28e-3** | 2.34e-2 ± 1.80e-3 | 2.18e-2 ± 1.65e-3 |
| Imagenet | MiX/L/16 | 4.30e-1 ± 9.59e-3 | 7.81e-3 ± 5.69e-4 | 1.13e-2 ± 8.20e-4 | 1.39e-2 ± 1.24e-3 | **7.03e-3 ± 6.30e-4** |
| Imagenet | BiT/d101/w1 | 2.50e-1 ± 6.00e-3 | 9.52e-3 ± 8.50e-4 | 4.77e-3 ± 3.07e-4 | 4.73e-3 ± 3.86e-4 | **4.53e-3 ± 3.54e-4** |
| Imagenet | BiT/d50/w1 | 2.17e-1 ± 5.36e-3 | 7.77e-3 ± 4.45e-4 | 4.40e-3 ± 2.47e-4 | 4.30e-3 ± 2.60e-4 | **3.64e-3 ± 2.56e-4** |
| Imagenet | ViT/B/16 | 3.69e-1 ± 8.98e-3 | 1.41e-2 ± 1.05e-3 | 1.04e-2 ± 9.16e-4 | 1.02e-2 ± 7.40e-4 | **7.68e-3 ± 7.60e-4** |
| Imagenet | MiX/B/16 | 3.21e-1 ± 8.18e-3 | 1.06e-2 ± 1.21e-3 | 7.86e-3 ± 4.83e-4 | **5.01e-3 ± 3.81e-4** | 5.78e-3 ± 5.70e-4 |
| Imagenet | BiT/d101/w3 | 3.26e-1 ± 8.10e-3 | **5.44e-3 ± 4.89e-4** | 1.00e-2 ± 8.45e-4 | 7.85e-3 ± 7.23e-4 | 7.37e-3 ± 7.56e-4 |
| Imagenet | BiT/d50/w3 | 3.09e-1 ± 8.07e-3 | 2.95e-2 ± 1.69e-3 | 6.19e-3 ± 3.91e-4 | 7.46e-3 ± 4.40e-4 | **6.03e-3 ± 3.93e-4** |

Table 2: Extrapolation Results for bivariate scaling behavior of downstream vision performance. See Section 4.1 for more details.

| Task | DC ↓ | A1 ↓ | A2 ↓ | A3 ↓ | UNSL ↓ |
|------|------|------|------|------|--------|
| Birds | 2.65e-1 ± 2.49e-2 | 7.38e-2 ± 1.41e-2 | 6.51e-2 ± 1.79e-2 | 4.77e-2 ± 7.51e-3 | **4.03e-2 ± 5.51e-3** |
| Imagenet | 2.54e-1 ± 2.02e-2 | 4.61e-2 ± 1.01e-2 | 3.39e-2 ± 1.06e-2 | 2.20e-2 ± 3.63e-3 | **1.70e-2 ± 2.76e-3** |

Table 3: Extrapolation Results for trivariate scaling behavior of downstream vision performance. See Section 4.1 for more details.

behavior, in Appendix 15.2.1 respectively see Figures 21, 22, 23, 24, 25; this trivariate scaling behavior data is from scaling behavior data released by Muennighoff et al. (2023), and the simultaneously varying dimensions of these trivariate scaling behaviors are number of model parameters, number of tokens processed, and number of tokens in training dataset. To view plots of UNSL, CF, A1, and A2 on each of these bivariate scaling behaviors, in Appendix 15.2.2 respectively see Figures 26, 27, 28, and 29; the simultaneously varying dimensions of these bivariate scaling behaviors are number of model parameters and number of training steps (or number of tokens processed). There is no A3 in Table 4 because UNSL becomes A3 in the scenario of Table 4, i.e. the scenario in which training dataset size is effectively infinite such that one only trains for one epoch. The bivariate scaling behaviors that are referred to as "constant" are obtained from the LLaMA and HGRN2 portions of Figures 1 and 2 of Shen et al. (2024); they are referred to as "constant" because the learning rate is held constant and a learning rate schedule is not used. The bivariate scaling behaviors that are referred to as "chinchilla" are obtained via correspondence with authors of Hoffmann et al. (2022); they are called "chinchilla" because they use "chinchilla-scaling" (i.e. a learning rate schedule that is chosen to be training compute optimal as in Hoffmann et al. (2022)) and are the scaling behavior data from Hoffmann et al. (2022). CSR (Common Sense Reasoning) is zero-shot mean test error rate across HellaSwag (Zellers et al., 2019), ARC (easy and challenge) (Clark et al., 2018), PIQA (Bisk et al., 2020), WinoGrande (Sakaguchi et al., 2020), OpenBookQA (Mihaylov et al., 2018), SIQA (Sap et al., 2019), and BoolQ (Clark et al., 2019).

| Task | Model | Scaling | CF ↓ | A1 ↓ | A2 ↓ | UNSL ↓ |
|------|-------|---------|------|------|------|--------|
| Upstream | Transformer | Chinchilla | 1.72e-2 ± 1.69e-3 | 9.85e-3 ± 1.30e-3 | 4.43e-3 ± 6.15e-4 | **3.81e-3 ± 6.52e-4** |
| LAMBADA | Transformer | Chinchilla | 2.08e-2 ± 2.48e-3 | 1.45e-2 ± 1.89e-3 | 1.30e-2 ± 1.80e-3 | **1.13e-2 ± 1.60e-3** |
| CSR | Transformer | Constant | 4.50e-2 ± 3.72e-3 | 1.43e-2 ± 1.24e-3 | 1.66e-2 ± 1.08e-3 | **1.28e-2 ± 1.05e-3** |
| LAMBADA | Transformer | Constant | 3.06e-2 ± 3.92e-3 | 4.17e-2 ± 3.52e-3 | 3.12e-2 ± 2.48e-3 | **2.24e-2 ± 1.71e-3** |
| Upstream | Transformer | Constant | 7.15e-2 ± 5.03e-3 | 3.98e-2 ± 3.27e-3 | 2.89e-2 ± 1.58e-3 | **7.95e-3 ± 6.63e-4** |
| CSR | Recurrent | Constant | 5.20e-2 ± 3.32e-3 | 2.65e-1 ± 2.87e-2 | **1.15e-2 ± 9.39e-4** | 1.22e-2 ± 9.15e-4 |
| LAMBADA | Recurrent | Constant | 3.02e-2 ± 2.63e-3 | 3.75e-2 ± 2.60e-3 | 4.31e-2 ± 3.62e-3 | **1.66e-2 ± 1.38e-3** |
| Upstream | Recurrent | Constant | 3.13e-2 ± 2.36e-3 | 3.07e-2 ± 1.99e-3 | 1.92e-2 ± 1.63e-3 | **4.66e-3 ± 3.51e-4** |

Table 4: Extrapolation Results for bivariate scaling behavior of downstream (and upstream) language performance. See Section 4.2 for more details.

| DC ↓ | A1 ↓ | A2 ↓ | A3 ↓ | UNSL ↓ |
|---|---|---|---|---|
| 6.24e-2 ± 6.00e-3 | 2.00e-2 ± 1.90e-3 | 1.96e-2 ± 3.62e-3 | 1.49e-02 ± 3.45e-03 | **7.82e-3 ± 1.33e-3** |

Table 5: Extrapolation Results for trivariate scaling behavior of language performance. See Section 4.2 for more details.

# 5    THE LIMIT OF THE PREDICTABILITY OF SCALING BEHAVIOR

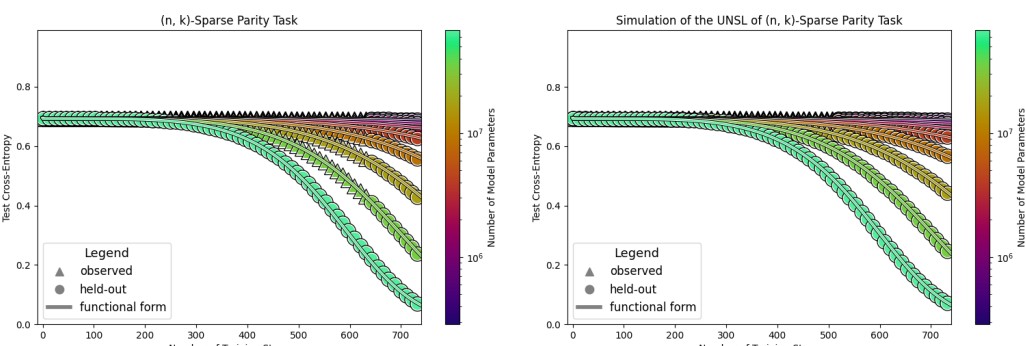

Figure 3: Extrapolation of UNSL on scaling behavior of an MLP trained for a single epoch on the (n, k)-sparse parity task (with $n = 40$ and $k = 4$) of Barak et al. (2022). Each point in the left plot is the mean of greater than 100 seeds. In the left plot, each point is gathered from an MLP trained for a single epoch on the (n, k)-sparse parity task (with $n = 40$ and $k = 4$) of Barak et al. (2022). In the right plot, each point is gathered from a noiseless simulation of the UNSL of the scaling behavior of that (n, k)-sparse parity task. See Section 5 and Appendix 11 for more details.

We use UNSL to glean insights about the limit of the predictability of scaling behavior. In Figure 3 left, UNSL accurately extrapolates the scaling behavior of the sparse parity task of Barak et al. (2022), despite the fact that this task famously does not exhibit any observable progress in loss (nor error) for the first few hundred training steps. In Figure 3 right, we use a noiseless simulation of the UNSL of the scaling behavior of the sparse parity task to show what would happen if one had infinitely many training runs / seeds to average out all the noisy deviation between runs such that one could recover (i.e. learn via curve-fitting) the learned constants of the UNSL as well as possible. We observe the following:

- To accurately extrapolate past each hyperbreak, the shortest distance to each hyperbreak from (the convex hull of) the points used for fitting must be sufficiently small.

# 6    DISCUSSION

We have presented the unified neural scaling law (UNSL) functional form that accurately models and extrapolates the scaling behaviors of deep neural networks as multiple dimensions all vary simultaneously (i.e. how the evaluation metric of interest varies as one simultaneously varies the number of model parameters, training dataset size, number of training steps, and various hyperparameters) for various architectures and for each of various tasks within a varied set of upstream and downstream tasks. When compared to other functional forms for neural scaling, this functional form yields extrapolations of scaling behavior that are considerably more accurate on this set.

APPENDIX

# 7 EXTENSION OF THE ADDITIVE SYMMETRY RELATIONS DISCUSSED IN SECTION 2.1 TO A SUMMATION OF TWO MBNSLS THAT EACH HAVE AN ARBITRARY NUMBER OF HYPERBREAKS $n$

Note that Equation 5 sums two $n = 0$ versions of MBNSL of Equation 4. To extend the relations discussed in Section 2.1 to a summation of two MBNSLs that each have an arbitrary number of hyperbreaks $n$, for each of those two MBNSLs one needs to obtain the $n = 0$ version $(w_b \prod_{i=1}^{m} x_i^{w_{c_i}})$ of MBNSL of Equation 4 that is the tangent hyperplane in multi-log space. The values of $w_b$ and $w_{c_i}$ that yield the tangent hyperplane in multi-log space are:

$$w_{c_i} = -c_{i_0} - \sum_{j=1}^{n} \text{sign}(f_j) \cdot c_{i_j} \cdot \left( 1 + \left( \frac{\prod_{l=1}^{m} x_l^{c_{l_j}}}{d_j} \right)^{-\left| \frac{1}{f_j} \right|} \right)^{-1},$$

$$w_b = b \cdot \left( \prod_{l=1}^{m} x_l^{-c_{l_0}} \right) \left( \prod_{j=1}^{n} \left( 1 + \left( \frac{\prod_{l=1}^{m} x_l^{c_{l_j}}}{d_j} \right)^{\left| \frac{1}{f_j} \right|} \right)^{-f_j} \right) \prod_{i=1}^{m} x_i^{-w_{c_i}}.$$

# 8 DEFINITION OF ROOT MEAN SQUARED LOG ERROR

$$Root\_Mean\_Squared\_Log\_Error = RMSLE = \sqrt{\frac{1}{N} \sum_{i=1}^{N} (log(y_i) - log(\hat{y}_i))^2}$$

# 9 DEFINITION OF ROOT STANDARD LOG ERROR

$$error = (log(y_i) - log(\hat{y}_i))^2)$$

$$\mu_{error} = \frac{1}{N} \sum_{i=1}^{N} error$$

$$\sigma_{error} = \sqrt{\frac{1}{N-1} \sum_{i=1}^{N} (error_i - \mu_{error})^2}$$

$$Root\_Standard\_Log\_Error = \sqrt{\mu_{error} + \frac{\sigma_{error}}{\sqrt{len(\hat{y})}}} - \sqrt{\mu_{error}}$$

# 10 EXPERIMENTAL DETAILS OF FITTING UNSL

We fit the UNSL by implementing it in KFAC-JAX (Botev & Martens, 2022) and minimizing mean squared log error (MSLE):

$$MSLE = \frac{1}{N} \sum_{i=1}^{N} (\log(y + \epsilon) - \log(\hat{y} + \epsilon))^2, \tag{7}$$

with $\epsilon = 10^{-16}$. We also employ L2 regularization on the exponents of the UNSL with a weighting of $\lambda$ relative to the MSLE loss term.

The values of $n$ (from Equation 4) (and $S$ from Equation 2) and $\lambda$ that yield the lowest extrapolation error can be obtained as follows. Split the set of observed points (i.e. the triangle shaped points) used for fitting into two sets, a validation set and a training set; for each of every point in the validation set, the training set should not contain a point that is simultaneously larger than each of every $x$ dimension ($x_1...x_m$) of that validation set point. The values of $n$, $S$, and $\lambda$ with the lowest validation error when fitting on the remaining training points are then used. Note that once the values of $n$, $S$, and $\lambda$ are identified, the validation set is added back to the training set; (and the hold-out points (i.e. the circle shaped points) are still held out to evaluate extrapolation RMSLE). In practice, $S$ is set equal to 1 unless the scaling behavior of interest is an extravagant scaling behavior that is similar to the scaling behavior shown in Figure 7 of Appendix 14.5.

It takes 20000 training steps and 20 seeds of random initialization for KFAC-JAX to converge when fitting a UNSL. We use the JAX default "LeCun Normal" initialization as the distribution from which each random initialization (for each seed) is drawn from for parameters of UNSL. Unlike the values of $n$ (from Equation 4) (and $S$ from Equation 2) and $\lambda$, the optimal seed that is selected is that which yields the lowest training error (not the lowest validation error).

## 11    EXPERIMENTAL DETAILS OF SECTIONS 5, 12, 14 (BESIDES FIGURE 8), AND 16

For all figures in Sections 5, 12, 14 (besides Figure 8), and 16:

- The batch size is 80000. No regularization is used because training dataset size is $\sim$infinite such that model is only trained for a single epoch. Adam is used. Adam hyperparameters are $\beta_1 = 0$ and $\beta_2 = 0$ (except for Figures 9 and 10 (and Table 6) in Section 14.7 in which $\beta_1 = 0.9$ and $\beta_2 = 0.999$). Except when learning rate and/or standard deviation of weights at initialization are explicitly varied in the plots of figures, learning rate and standard deviation of weights at initialization are held constant.

In Figure 3, number of model parameters is varied by varying width when depth is held constant.

## 12    EFFECT OF VARYING THE NUMBER OF OBSERVED POINTS USED FOR FITTING UNSL FUNCTIONAL FORM

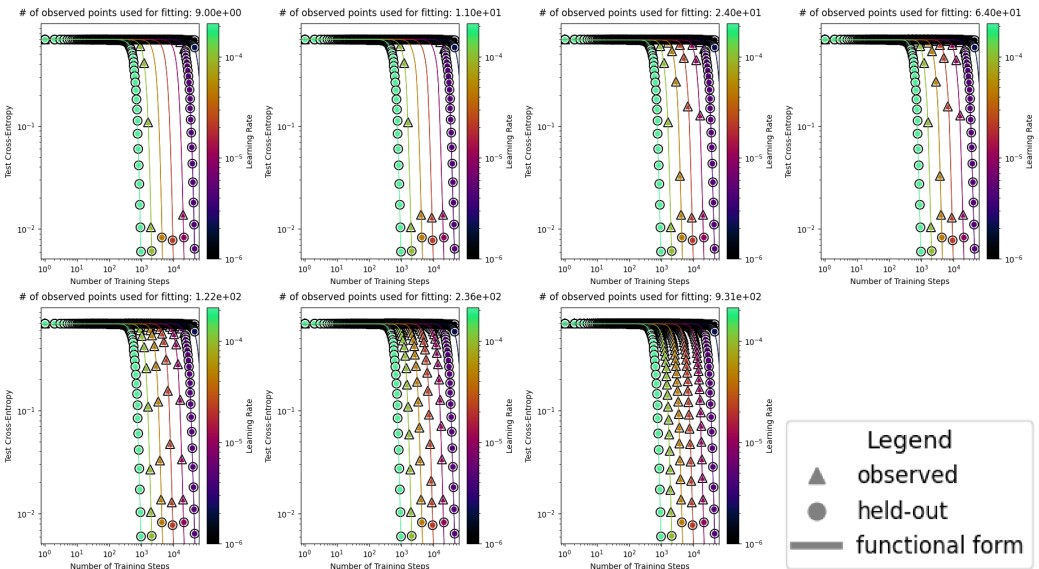

Figure 4: Varying the number of observed points used for fitting UNSL functional form from $9e0$ (in top left plot) to $9e2$ (in bottom right plot). Scaling behavior is that of an MLP trained for a single epoch on the (n, k)-sparse parity task (with $n = 40$ and $k = 4$) of Barak et al. (2022). See Appendix 12 for more details.

In Figure 4, we observe that UNSL accurately extrapolates scaling behavior when only a small number of observed points are used for fitting UNSL functional form.

## 13 EXPLANATION OF HOW UNSL FUNCTIONAL FORM SATISFIES ALL OF THE DESIDERATA OF SECTION 2.2

### 13.1 EXPLANATION OF HOW UNSL FUNCTIONAL FORM SATISFIES DESIDERATUM 1

Desideratum 1 says that for each single input dimension $x_i$, the scaling behavior follows a univariate broken neural scaling law of Caballero et al. (2023), i.e.:

$$ y = b \cdot x_i^{-c_{i_0}} \prod_{j=1}^{n} \left( 1 + \left( \frac{x_i^{c_{ij}}}{d_j} \right)^{\left| \frac{1}{f_j} \right|} \right)^{-f_j}, $$

where $b$, $c_{i_0}$, $c_{i1}...c_{ij}$, $d_1...d_j$, and $f_1...f_j$ are learned parameters. (Note that "performance limit" term $a$ from Caballero et al. (2023) is intentionally removed here because it is addressed by other desiderata.)

This is implemented in Equation 4, where each univariate scaling behavior is modeled as a Broken Neural Scaling Law (BNSL):

$$ K(M, n, k) = b_k \cdot \left( \prod_{i \in M} x_i^{-c_{i_{0_k}}} \right) \prod_{j=1}^{n} \left( 1 + \left( \frac{\prod_{i \in M} x_i^{c_{i_{j_k}}}}{d_{j_k}} \right)^{\left| \frac{1}{f_{j_k}} \right|} \right)^{-f_{j_k}}. $$

For the pedagogical purposes of this Section 13.1, by setting $M = \{1, \ldots, m\}$ and removing subscript $k$ one can simplify Equation 4 to:

$$ y = b \cdot \left( \prod_{i=1}^{m} x_i^{-c_{i_0}} \right) \prod_{j=1}^{n} \left( 1 + \left( \frac{\prod_{i=1}^{m} x_i^{c_{ij}}}{d_j} \right)^{\left| \frac{1}{f_j} \right|} \right)^{-f_j}. $$

In that equation, when one varies only a single input dimension $x_i$, all $x$ with a subscript in "$\{1, \ldots, m\} \setminus \{i\}$" become constants that can be folded into $b$ or $d_j$, hence recovering the univariate broken neural scaling law of Caballero et al. (2023), i.e.:

$$y = b \cdot x_i^{-c_{i_0}} \prod_{j=1}^{n} \left( 1 + \left( \frac{x_i^{c_{i_j}}}{d_j} \right)^{\left| \frac{1}{f_j} \right|} \right)^{-f_j},$$

### 13.2   EXPLANATION OF HOW UNSL FUNCTIONAL FORM SATISFIES DESIDERATUM 2

In Equation 4, the j-th hyperbreak (i.e. smooth transition from the j-th hyperplane to the (j+1)-th hyperplane) occurs at the values of $(x_i)_{i \in M}$ for which this equality is true:

$$d_{j_k} = \prod_{i \in M} x_i^{c_{i_{j_k}}}.$$

As can be seen from this equality, the location at which each hyperbreak occurs is shifted via multiplicative interaction between (exponentiations of) input dimensions $(x_i)_{i \in M}$.

### 13.3   EXPLANATION OF HOW UNSL FUNCTIONAL FORM SATISFIES DESIDERATUM 3

For the pedagogical purposes of this Section 13.3, by removing subscript $k$ one can simplify Equation 4 to:

$$y = b \cdot \left( \prod_{i \in M} x_i^{-c_{i_0}} \right) \prod_{j=1}^{n} \left( 1 + \left( \frac{\prod_{i \in M} x_i^{c_{i_j}}}{d_j} \right)^{\left| \frac{1}{f_j} \right|} \right)^{-f_j}.$$

When $c_{i_j}$ and $f_j$ are constrained to force that functional form to always be nonmonotonic (and assuming $x_i > 0$, $y > 0$, $b > 0$, $d_j > 0$), that functional form effectively becomes the following monomial when maximally close to the global minima with respect to $y$:

$$y = \left( b \cdot \prod_{\substack{j=1 \\ f_j > 0}}^{n} d_j \right) \cdot \prod_{i \in M} x_i^{-\left( c_{i_0} + \sum_{\substack{j=1 \\ f_j > 0}}^{n} c_{i_j} \right)}.$$

When Equation 4 becomes a monomial, the expressivity of Equation 3 becomes equivalent to the expressivity of this functional form:

$$b \cdot \left( \prod_{i \in U_r} x_i^{-c_{i_{0_0}}} \right) + \sum_{t \in T_r} b_t \, x_t^{-c_{t_{0_t}}},$$

which effectively becomes

$$\sum_{t \in T_r} b_t \, x_t^{-c_{t_{0_t}}}$$

when

$$b \prod_{i \in U_r} x_i^{-c_{i_{0_0}}} \ll \sum_{t \in T_r} b_t \, x_t^{-c_{t_{0_t}}}.$$

As a result, it is also true that

$$a \;+\; b \cdot \left( \prod_{i \in U_r} x_i^{-c_{i_{0_0}}} \right) \;+\; \sum_{t \in T_r} b_t \, x_t^{-c_{t_{0_t}}}$$

effectively becomes

$$a \;+\; \sum_{t \in T_r} b_t \, x_t^{-c_{t_{0_t}}}$$

when

$$b \prod_{i \in U_r} x_i^{-c_{i_{0_0}}} \;\ll\; \sum_{t \in T_r} b_t \, x_t^{-c_{t_{0_t}}} \,.$$

Additionally, the following functional form

$$a \;+\; \sum_{t \in T_r} b_t \, x_t^{-c_{t_{0_t}}}$$

effectively becomes

$$a \;+\; b_v \, x_v^{-c_{v_{0_v}}}, \quad (\text{where } v \in T_r)$$

when

$$\left( a + \sum_{t \in T_r \setminus \{v\}} b_t \, x_t^{-c_{t_{0_t}}} \right) \;\ll\; \left( a + b_v \, x_v^{-c_{v_{0_v}}} \right).$$

### 13.4 EXPLANATION OF HOW UNSL FUNCTIONAL FORM SATISFIES DESIDERATUM 4

Recall that Equation 1 is:

$$y = (a_0 + Q(2)) + \left( Q(S+3) + a_1^{-1} \right)^{-1}.$$

For pedagogical purposes, this equation can be rewritten as:

$$y = a_0 + \left( Q(2) + \left( Q(S+3) + a_1^{-1} \right)^{-1} \right).$$

Desideratum 4 is captured by the addition between $a_0$ and $\left( Q(2) + \left( Q(S+3) + a_1^{-1} \right)^{-1} \right)$ in this equation, where $a_0$ represents the ultimate limit of performance.

### 13.5 EXPLANATION OF HOW UNSL FUNCTIONAL FORM SATISFIES DESIDERATUM 5

This is captured in Equations 1 and 2 when (the reciprocal of) each of every variable in the set $\{a_i\}_{i>0}$ is summed with a functional form and each resultant sum is then raised to the $-1$ power. The set $\{a_i\}_{i>0}$ contains multiple variables rather than a single variable because misperformance caused by different phenomena often have different misperformance limits. For example, misperformance caused by overfitting often has a misperformance limit that is significantly worse than the performance of random guessing; meanwhile, misperformance caused by nonoptimal hyperparameters often has at least one misperformance limit that is equal to the performance of random guessing. The reason that in Equation 2 a value of $S$ greater than 1 (rather than equal to 1) is sometimes used is that there sometimes are multiple misperformance limits $a_{q+s}$ (e.g. as in Figure 7 of Appendix 14.5): a

misperformance limit that is significantly larger than random guessing (that usually is noticeable when the number of training steps is small) and a misperformance limit that approximately is less than or equal to random guessing (that usually is noticeable when the number of training steps is large).

### 13.6 EXPLANATION OF HOW UNSL FUNCTIONAL FORM SATISFIES DESIDERATUM 6

Recall Appendix 13.3. As a result of Appendix 13.3, Desideratum 6 is captured by each of every instance in which $R(r)$ of Equation 3 is effectively raised to the $-1$ power; an instance in which $R(r)$ occurs is considered "effectively raised to the $-1$ power" if the count of reciprocal operations whose scope contains that instance is odd. Instances in which this occurs are $\left(R(q+s) + a_{q+s}{}^{-1}\right)^{-1}$ from Equation 2 and $\left(Q(S+3) + a_1{}^{-1}\right)^{-1}$ from Equation 1 (which contains $\left((R(q))^{-1} + a_q{}^{-1}\right)^{-1}$ from Equation 2).

### 13.7 EXPLANATION OF HOW UNSL FUNCTIONAL FORM SATISFIES DESIDERATUM 7

This desideratum is captured when $\left((R(q))^{-1} + a_q{}^{-1}\right)^{-1}$ is summed with the "oppositional force of hyperparameters" in Equation 2, and when $(a_0 + Q(2))$ is summed with the "oppositional force of overfitting" in Equation 1.

### 13.8 EXPLANATION OF HOW UNSL FUNCTIONAL FORM SATISFIES DESIDERATUM 8

UNSL (i.e. Equation 1) functional form (expanded out for pedagogical purposes) is:

$$
y = \left( a_0 + \left((R(2))^{-1} + a_2{}^{-1}\right)^{-1} + \underbrace{\sum_{s=1}^{S} \left(R(2+s) + a_{2+s}{}^{-1}\right)^{-1}}_{\text{oppositional force of hyperparameters}} \right)
$$

$$
+ \underbrace{\left( a_1{}^{-1} + \left((R(S+3))^{-1} + a_{S+3}{}^{-1}\right)^{-1} + \underbrace{\sum_{s=1}^{S} \left(R(S+3+s) + a_{S+3+s}{}^{-1}\right)^{-1}}_{\text{oppositional force of hyperparameters}} \right)^{-1}}_{\text{oppositional force of overfitting}}.
$$

As can be seen in that expansion of UNSL, oppositional force(s) of hyperparameters oppose the "oppositional force of overfitting" and the subset of the UNSL functional form that is not the "oppositional force of overfitting". Note that each of every "oppositional force" is nonnegative and that what each of every "oppositional force" opposes is nonnegative.

## 14 EMPIRICAL EVIDENCE OF DESIDERATA OF SECTION 2.2

### 14.1 EMPIRICAL EVIDENCE OF DESIDERATUM 1

In Figure 5, Desideratum 1 is true empirically. As can be seen in Figure 5, the sharpness needs to be decoupled from the amount of change in gradient (i.e. via the extra expressivity granted by $f$ in Equation 6 (and Equation 4)) in order to accurately fit and accurately extrapolate the scaling behavior.

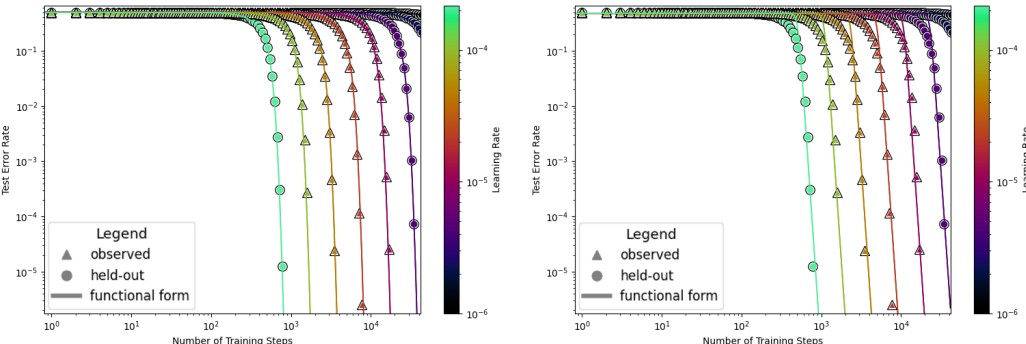

Figure 5: Extrapolation Results on scaling behavior of an MLP trained for a single epoch on the (n, k)-sparse parity task (with $n = 40$ and $k = 4$) of Barak et al. (2022). Left figure fits the functional

form $y = \left( \left( \left( b \prod_{i=1}^{m} x_i^{-c_{i_0}} \right) \left( 1 + \left( \frac{\prod_{i=1}^{m} x_i^{c_{i_1}}}{d} \right)^{\left| \frac{1}{f} \right|} \right)^{-f} \right)^{-1} + a^{-1} \right)^{-1}$ . Right figure fits the

functional form of left figure when $f$ is constrained to be 1 such that the functional form of right

figure is $\quad y = \left( \left( \left( b \prod_{i=1}^{m} x_i^{-c_{i_0}} \right) \left( 1 + \left( \frac{\prod_{i=1}^{m} x_i^{c_{i_1}}}{d} \right)^{\left| \frac{1}{1} \right|} \right)^{-1} \right)^{-1} + a^{-1} \right)^{-1}$ . Observe that the

fits and extrapolations in the top right quadrant of right figure are unsatisfactory. See Section 14.1 for more details.

## 14.2 EMPIRICAL EVIDENCE OF DESIDERATUM 2

Recall that in Equation 4 (with subscript $k$ removed for pedagogical purposes) the j-th hyperbreak (i.e. smooth transition from the j-th hyperplane to the (j+1)-th hyperplane) occurs at the values of $(x_i)_{i \in M}$ for which this equality is true:

$$d_j = \prod_{i \in M} x_i^{c_{i_j}} .$$

As a result, desideratum 2 is true empirically because the functional form $y =$

$\left( \left( \left( b \prod_{i=1}^{m} x_i^{-c_{i_0}} \right) \left( 1 + \left( \frac{\prod_{i=1}^{m} x_i^{c_{i_1}}}{d} \right)^{\left| \frac{1}{f} \right|} \right)^{-f} \right)^{-1} + a^{-1} \right)^{-1}$ accurately fits and accurately ex-

trapolates the scaling behavior in Figure 5 left.

## 14.3 EMPIRICAL EVIDENCE OF DESIDERATUM 3

Note that $(x_i)_{i=1}^{m} \in \overline{\mathbb{R}}_{>0}^{m}$ and that $(x_t)_{t \in T_r} \in \overline{\mathbb{R}}_{>0}^{T_r}$.

Desideratum 3 is observed empirically in several prior works such as Hoffmann et al. (2022) which empirically show that the scaling behavior follows $y = a + \sum_{t \in T_r} b_t \cdot x_t^{-c_t}$ when sufficiently close to the global optimum of $y$.

## 14.4 EMPIRICAL EVIDENCE OF DESIDERATUM 4

Desideratum 4 is observed empirically in several prior works such as Hoffmann et al. (2022) in which the transition to the performance limit as all $x_i$ dimensions in $x_1...x_m$ simultaneously go to the values of $(x_i)_{i=1}^{m} \in \overline{\mathbb{R}}_{>0}^{m}$ that yield the global optimum of $y$ is characterized by summing an entire functional form with a constant.

## 14.5 EMPIRICAL EVIDENCE OF DESIDERATUM 5

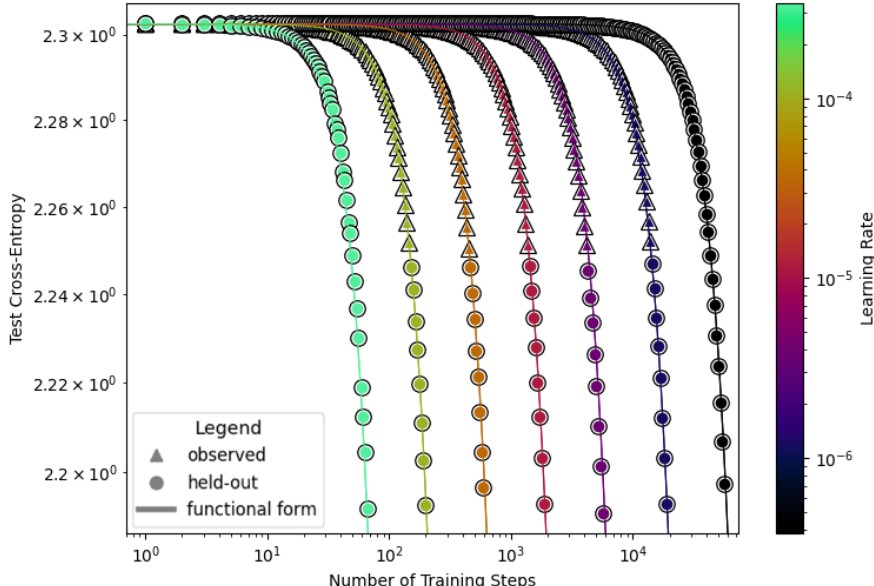

Figure 6: Extrapolation Results of functional form $y = \left( \left( b \prod_{i=1}^{m} x_i^{-c_i} \right)^{-1} + a^{-1} \right)^{-1}$. Scaling behavior is (top left region of) that of an MLP trained for a single epoch on dataset of Greydanus & Kobak (2024). See Section 14.5 for more details.

In Figure 6, Desideratum 5 is true empirically. As can be seen in Figure 6, the scaling behavior is characterized by raising to the -1 power the sum of a functional form and a constant.

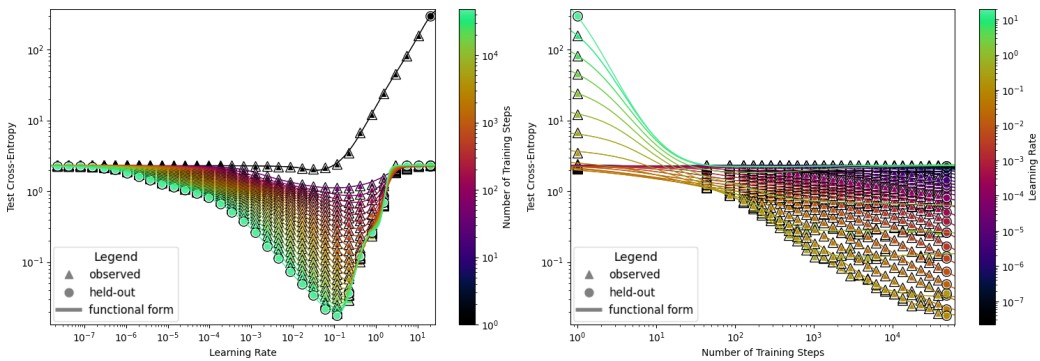

Figure 7: Extrapolation Results of UNSL functional form. Scaling behavior is that of an MLP (when standard deviation of weights at initialization is large) trained for a single epoch on dataset of Greydanus & Kobak (2024). See Section 14.5 for more details.

In Figure 7 with regards to Desideratum 5, there is a misperformance limit equal to random guessing performance (cross-entropy of 2.3) when it is simultaneously true that learning rate is large (i.e. greater than 3) and number of training steps is large (i.e. greater than 100); and an additional misperformance limit equal to a value significantly larger (i.e. larger than the largest $y$-axis value of Figure 7) than random guessing performance (cross-entropy of 2.3) occurs when it is simultaneously true that learning rate is large (i.e. significantly greater than 20) and number of training steps is small (i.e. less than 2). As a result, $S$ (from Equation 2) is equal to 2 in Figure 7.

## 14.6 EMPIRICAL EVIDENCE OF DESIDERATUM 6

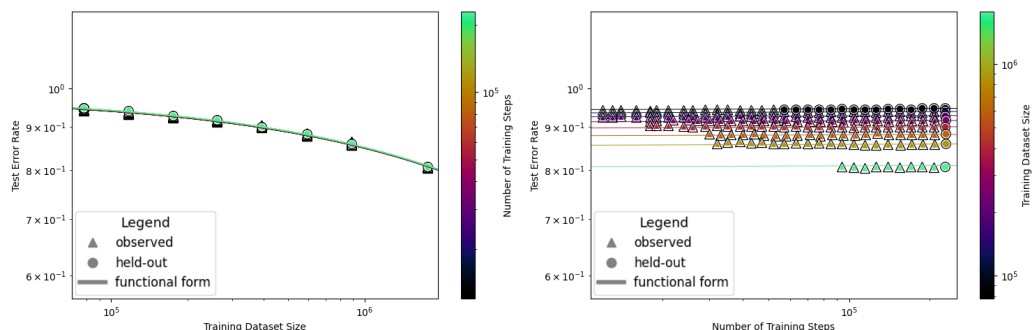

Figure 8: Extrapolation Results of functional form $y = \left(a + \sum_{t \in T_r} b_t \cdot x_t^{-c_t}\right)^{-1}$. Scaling behavior is (top region of the scaling behavior of) downstream ImageNet test error rate of ViT pre-trained on JFT. See Sections 14.6 and 4.1 for more details.

In Figure 8, Desideratum 6 is true empirically. As can be seen in Figure 8, the scaling behavior is characterized by the functional form $y = \left(a + \sum_{t \in T_r} b_t \cdot x_t^{-c_t}\right)^{-1}$.

### 14.7 EMPIRICAL EVIDENCE OF DESIDERATUM 7

In Table 6 which summarizes Figures 9 and 10, Desideratum 7 is true empirically. We obtain the trivariate scaling behavior as learning rate, standard deviation of weights at initialization, and number of training steps vary when training an MLP for a single epoch on dataset of Greydanus & Kobak (2024). When holding the number of learned parameters of the functional forms constant, we compare the training and extrapolation RMSLE of UNSL to the following ablated functional form baseline in which the additive symmetries of Equation 2 are removed:

$$y = a_0 + K(U_0, \ n_{0_0}, \ 0) + \sum_{t \in T_0} K(\{t\}, \ n_{0_t}, \ t), \quad \text{where } U_0, T_0 \subseteq \{1, \ldots, m\}. \tag{8}$$

As can be seen in Table 6 and Figures 9 and 10, when holding the number of learned parameters of the functional forms constant, UNSL yields fits and extrapolations with lower RMSLE than the ablated functional form baseline of Equation 8.

| Set | Baseline ↓ | UNSL ↓ |
|---|---|---|
| Training | 3.80e-2 ± 1.14e-3 | **3.49e-2 ± 1.27e-3** |
| Extrapolation | 8.09e-2 ± 5.90e-3 | **5.11e-2 ± 4.30e-3** |
| All | 4.14e-2 ± 1.26e-3 | **3.60e-2 ± 1.23e-3** |

Table 6: Results on trivariate scaling behavior in which Desideratum 7 is true empirically. See Section 14.7 for more details.

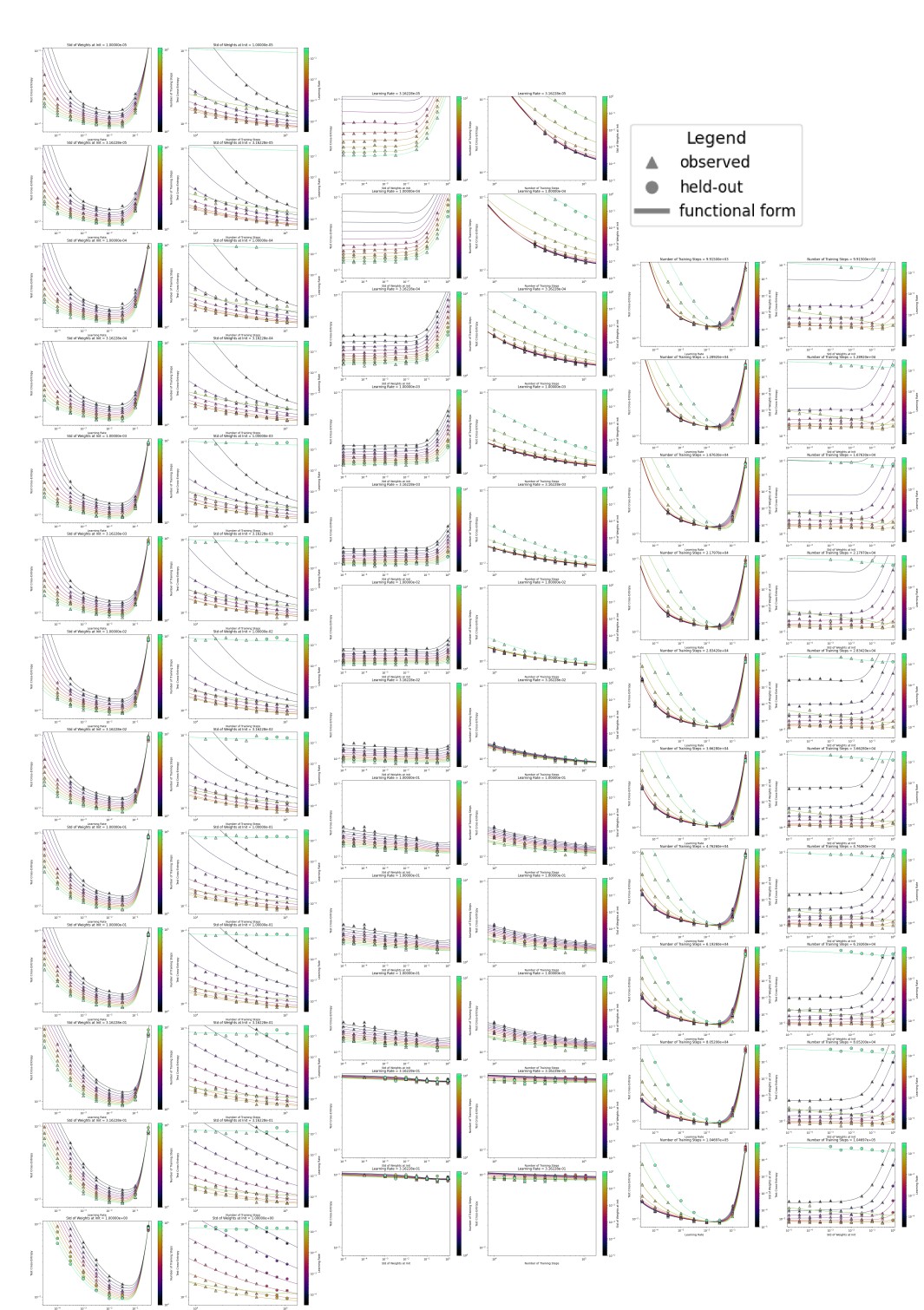

Figure 9: Extrapolation Results of UNSL. This trivariate scaling behavior is that of an MLP trained for a single epoch on dataset of Greydanus & Kobak (2024). See Section 14.7 for more details.

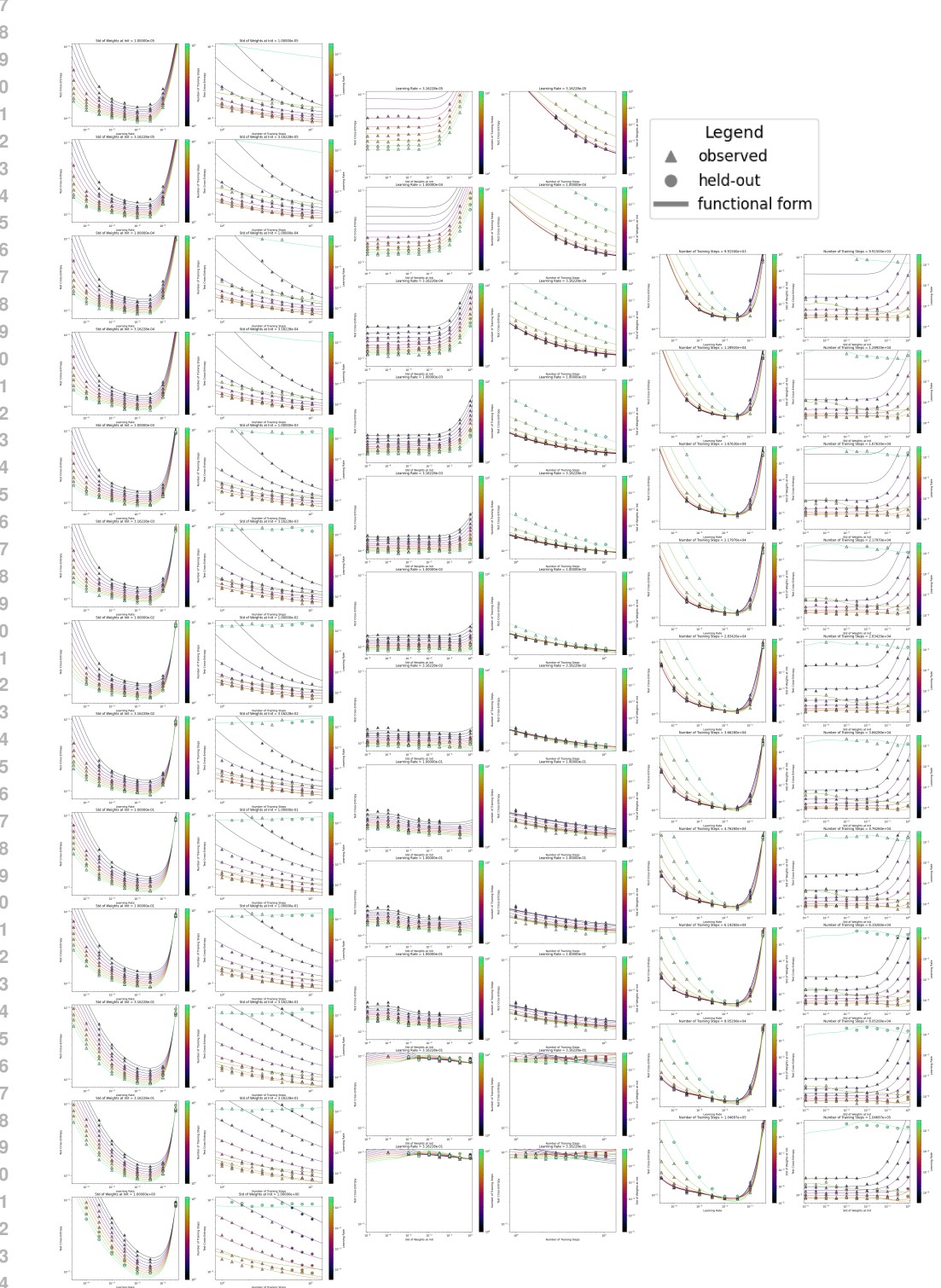

Figure 10: Extrapolation Results of ablation baseline of Equation 8. This trivariate scaling behavior is that of an MLP trained for a single epoch on dataset of Greydanus & Kobak (2024). See Section 14.7 for more details.

## 14.8 EMPIRICAL EVIDENCE OF DESIDERATUM 8

In Table 2, Desideratum 8 is true empirically because UNSL functional form outperforms A3 in the majority of instances.

Recall that A3 functional form is:

$$y = a_0 + \left((R(0))^{-1} + a_1^{-1}\right)^{-1} + \underbrace{\sum_{s=1}^{S} \left(R(s) + a_{s+1}^{-1}\right)^{-1}}_{\text{all oppositional forces in general}}.$$

meanwhile UNSL functional form (expanded out for pedagogical purposes) is:

$$y = \left( a_0 + \left((R(2))^{-1} + a_2^{-1}\right)^{-1} + \underbrace{\sum_{s=1}^{S} \left(R(2+s) + a_{2+s}^{-1}\right)^{-1}}_{\text{oppositional force of hyperparameters}} \right)$$

$$+ \underbrace{\left( a_1^{-1} + \left((R(S+3))^{-1} + a_{S+3}^{-1}\right)^{-1} + \underbrace{\sum_{s=1}^{S} \left(R(S+3+s) + a_{S+3+s}^{-1}\right)^{-1}}_{\text{oppositional force of hyperparameters}} \right)^{-1}}_{\text{oppositional force of overfitting}}.$$

As can be seen in that expansion of UNSL, oppositional force(s) of hyperparameters oppose the "oppositional force of overfitting" and the subset of the UNSL functional form that is not the "oppositional force of overfitting"; meanwhile, in A3 functional form, the "oppositional force(s) of hyperparameters" does not oppose the "oppositional force of overfitting". Note that each of every "oppositional force" is nonnegative and that what each of every "oppositional force" opposes is nonnegative.

## 15 PLOTS OF EXTRAPOLATION RESULTS

### 15.1 PLOTS OF VISION EXTRAPOLATION RESULTS

#### 15.1.1 BIVARIATE

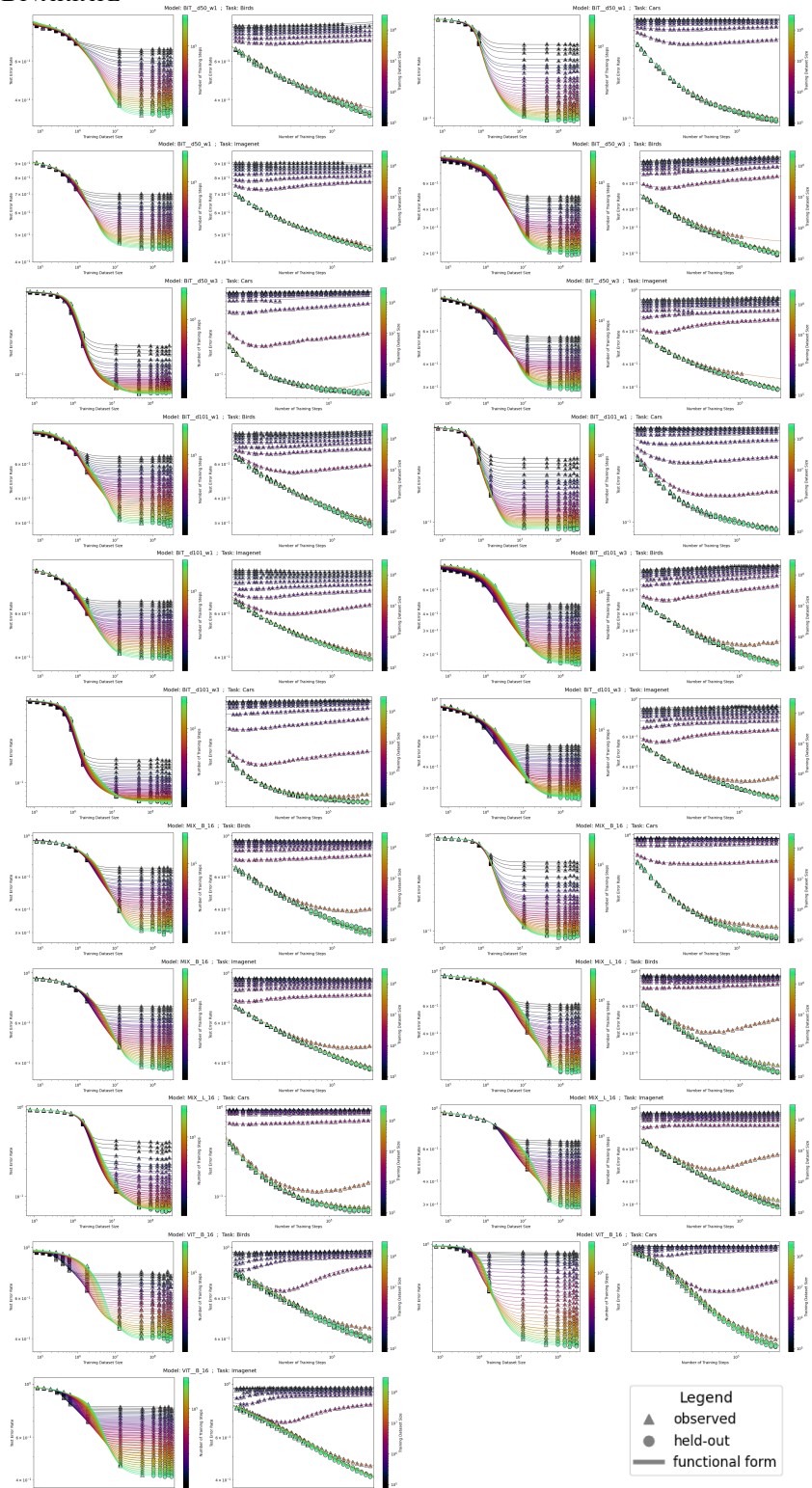

Figure 11: Extrapolation Results of UNSL on bivariate scaling behavior of downstream vision performance. See Section 4.1 for more details.

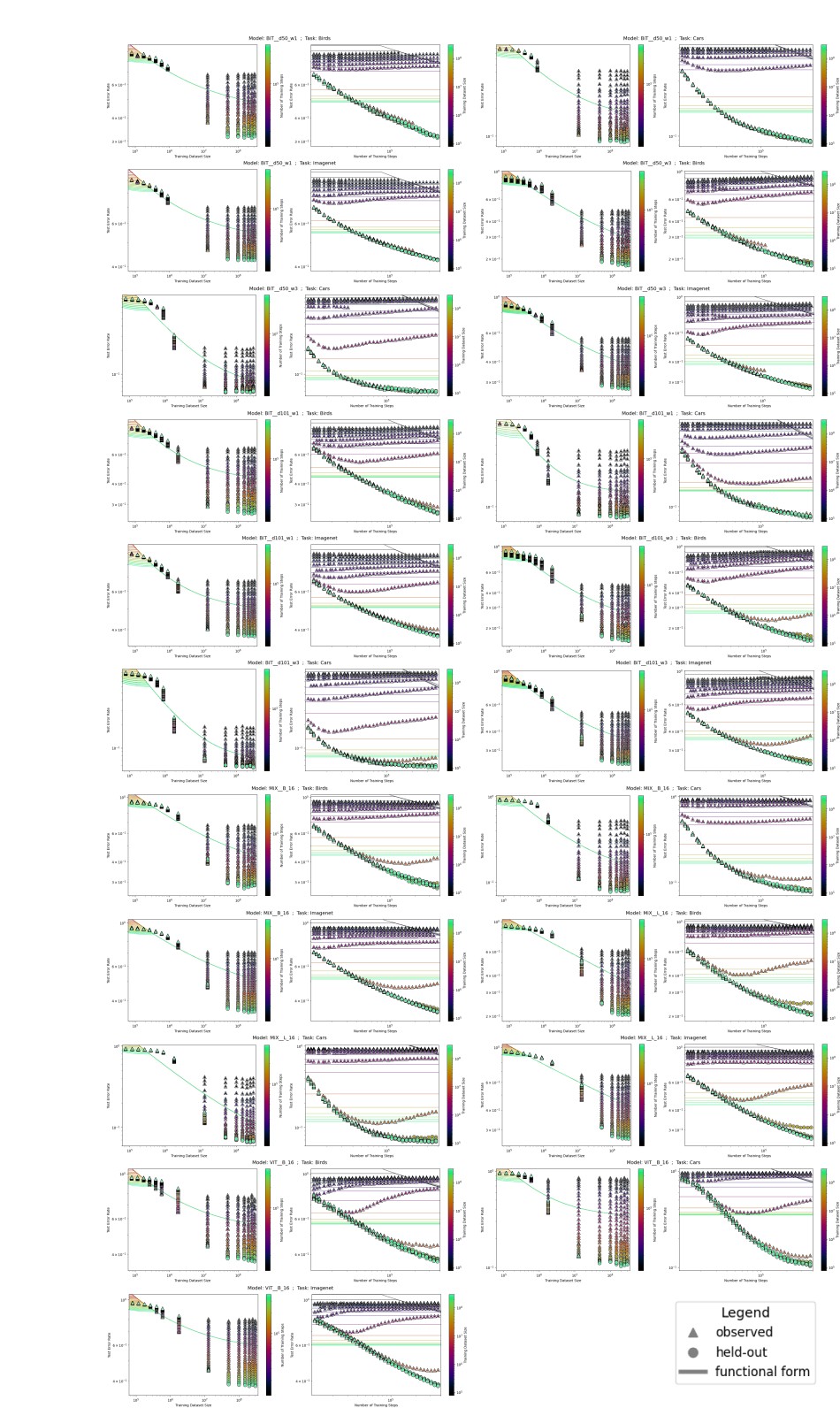

Figure 12: Extrapolation Results of "DC" functional form of Muennighoff et al. (2023) on bivariate scaling behavior of downstream vision performance. See Section 4.1 for more details.

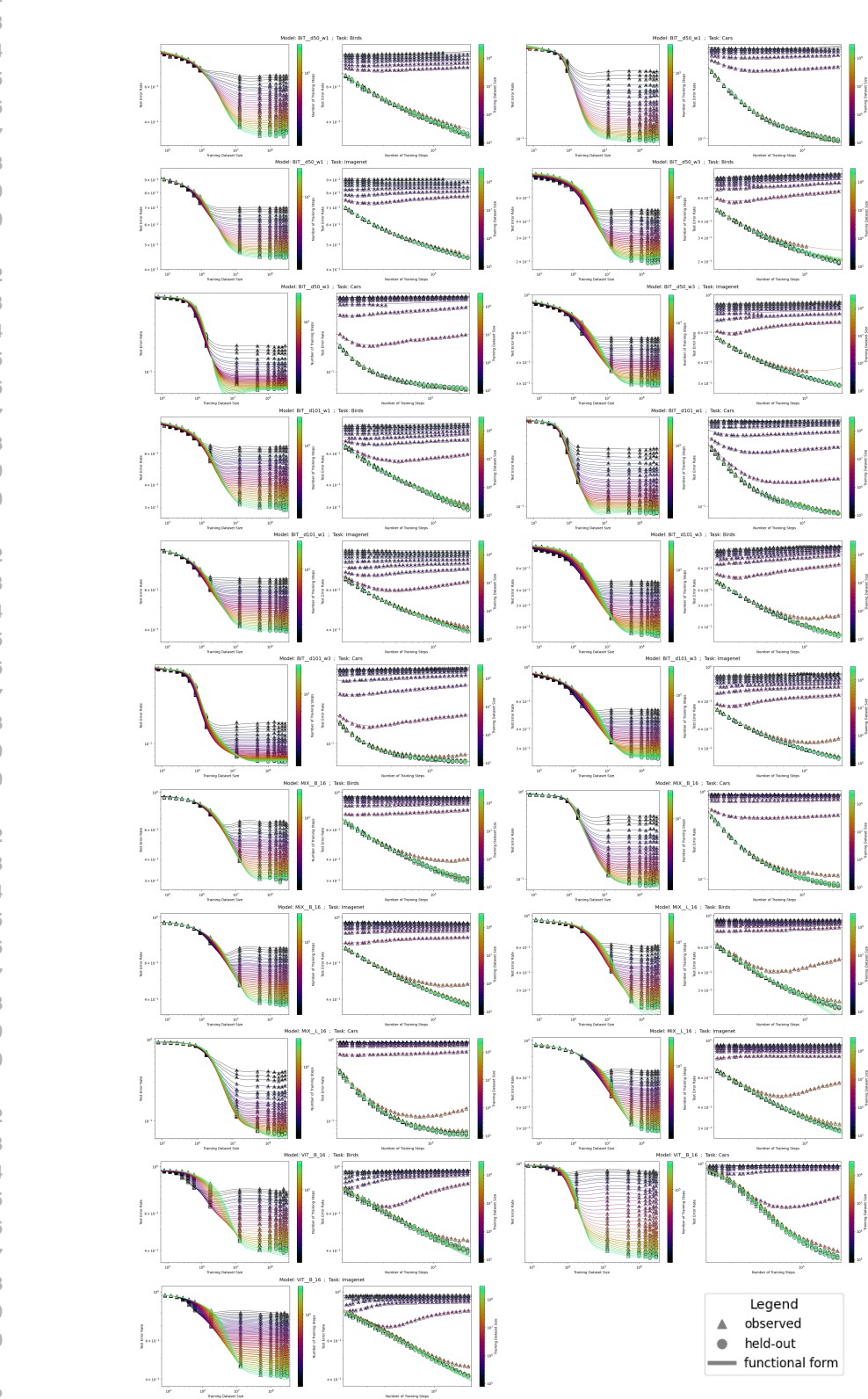

Figure 13: Extrapolation Results of A1 functional form on bivariate scaling behavior of downstream vision performance. See Section 4.1 for more details.

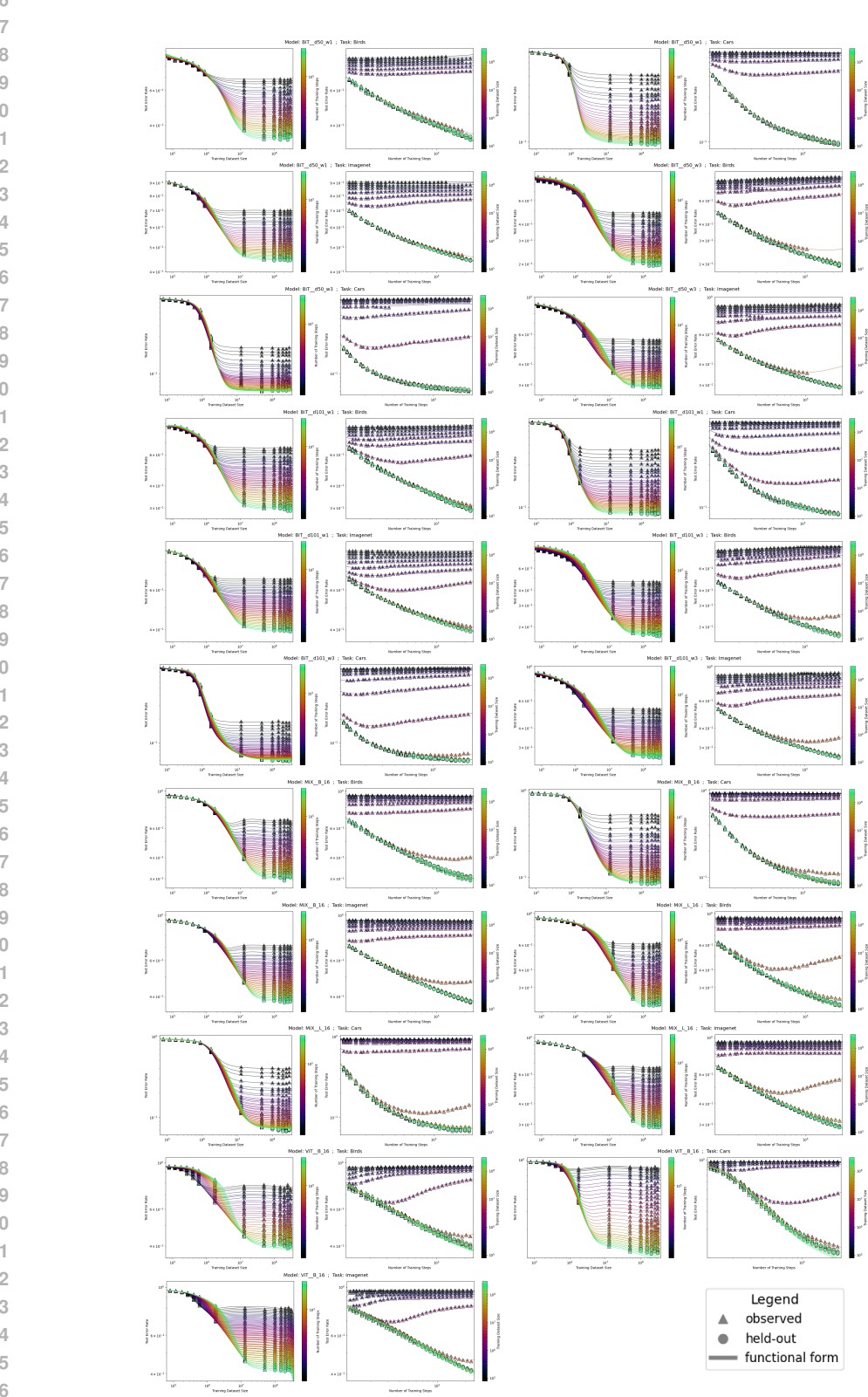

Figure 14: Extrapolation Results of A2 functional form on bivariate scaling behavior of downstream vision performance. See Section 4.1 for more details.

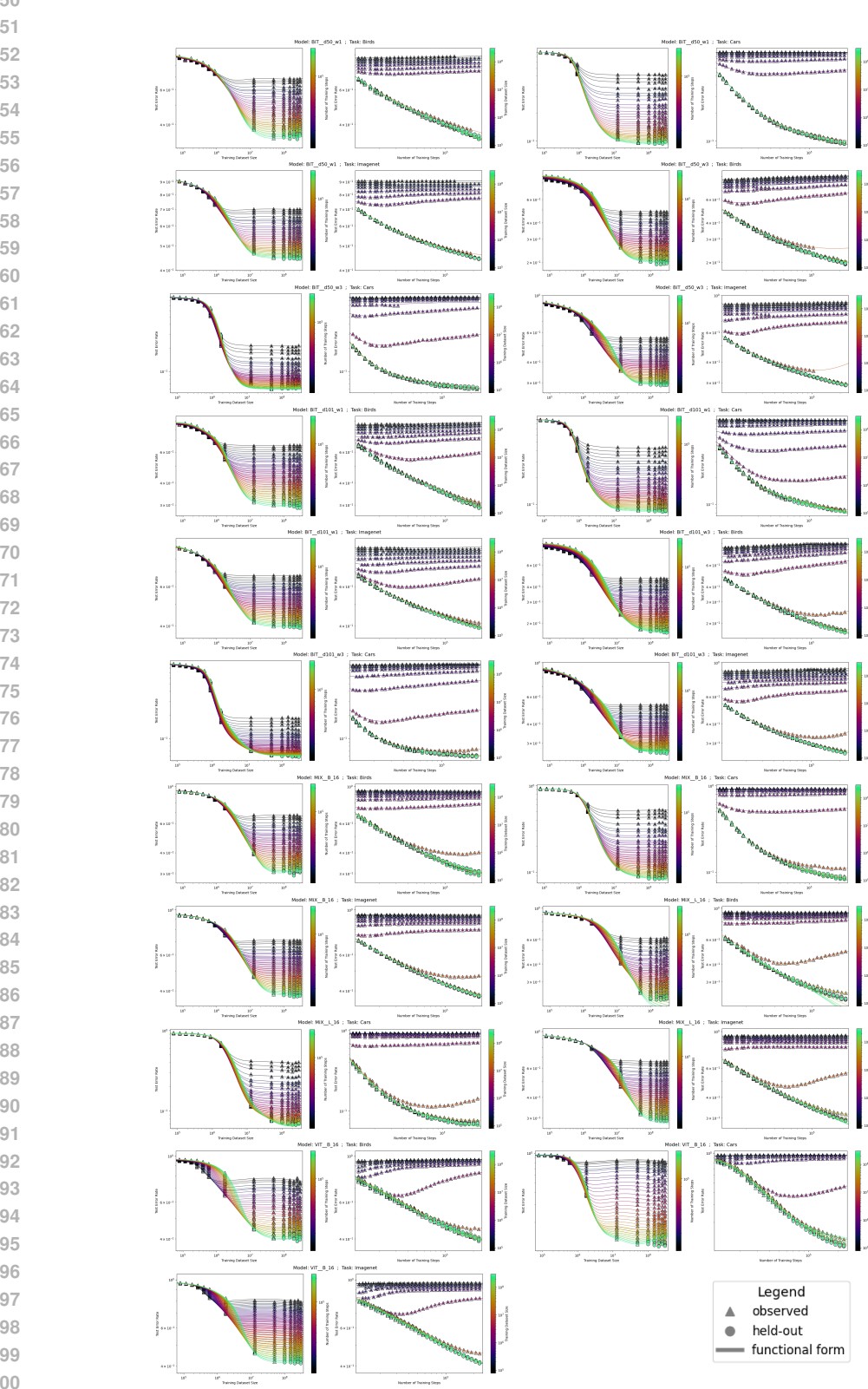

Figure 15: Extrapolation Results of A3 functional form on bivariate scaling behavior of downstream vision performance. See Section 4.1 for more details.

### 15.1.2 TRIVARIATE

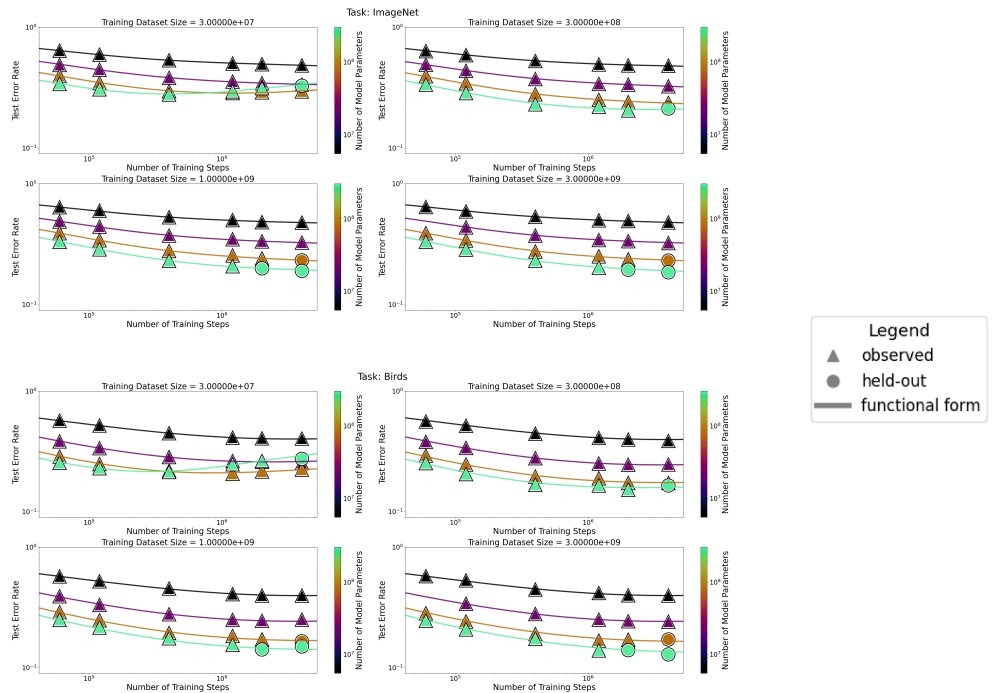

Figure 16: Extrapolation Results of UNSL functional form on trivariate scaling behavior of downstream vision performance. See Section 4.1 for more details.

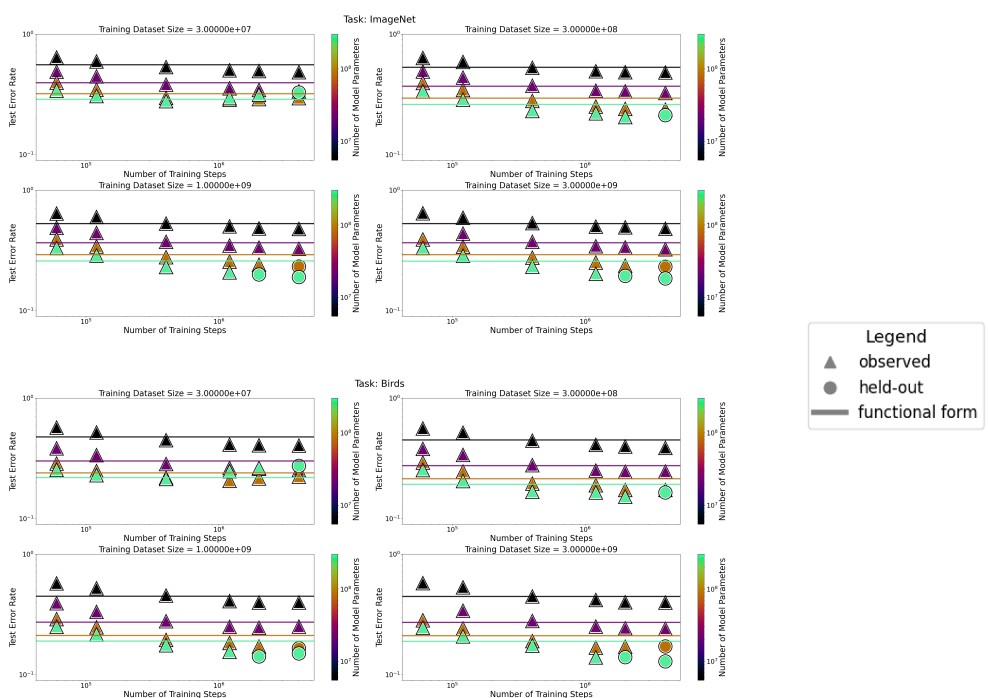

Figure 17: Extrapolation Results of "DC" functional form of Muennighoff et al. (2023) on trivariate scaling behavior of downstream vision performance. See Section 4.1 for more details.

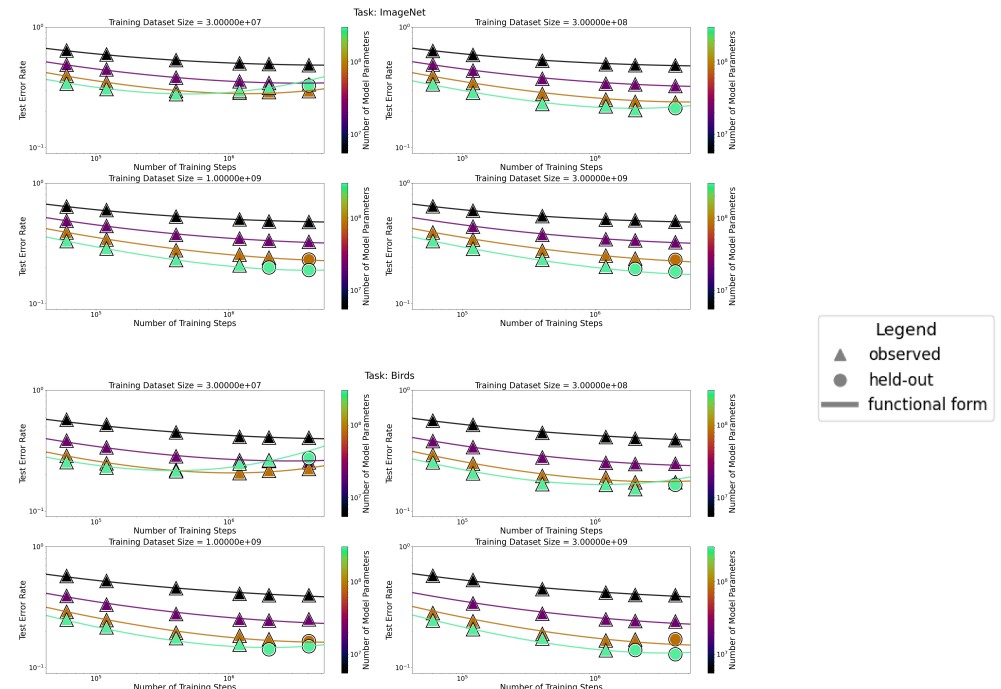

Figure 18: Extrapolation Results of A1 functional form on trivariate scaling behavior of downstream vision performance. See Section 4.1 for more details.

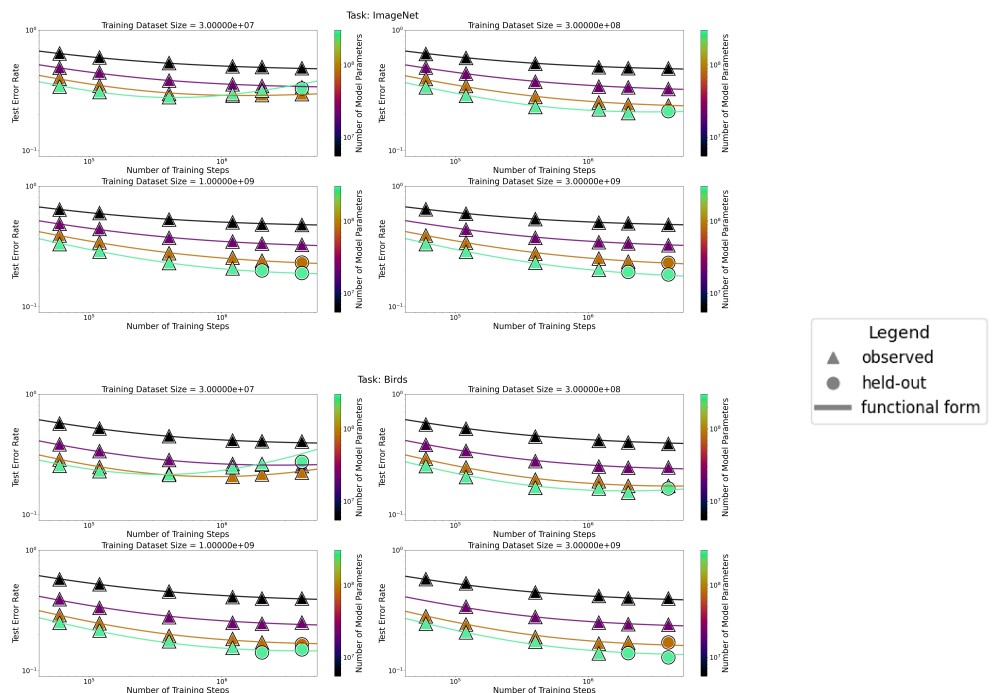

Figure 19: Extrapolation Results of A2 functional form on trivariate scaling behavior of downstream vision performance. See Section 4.1 for more details.

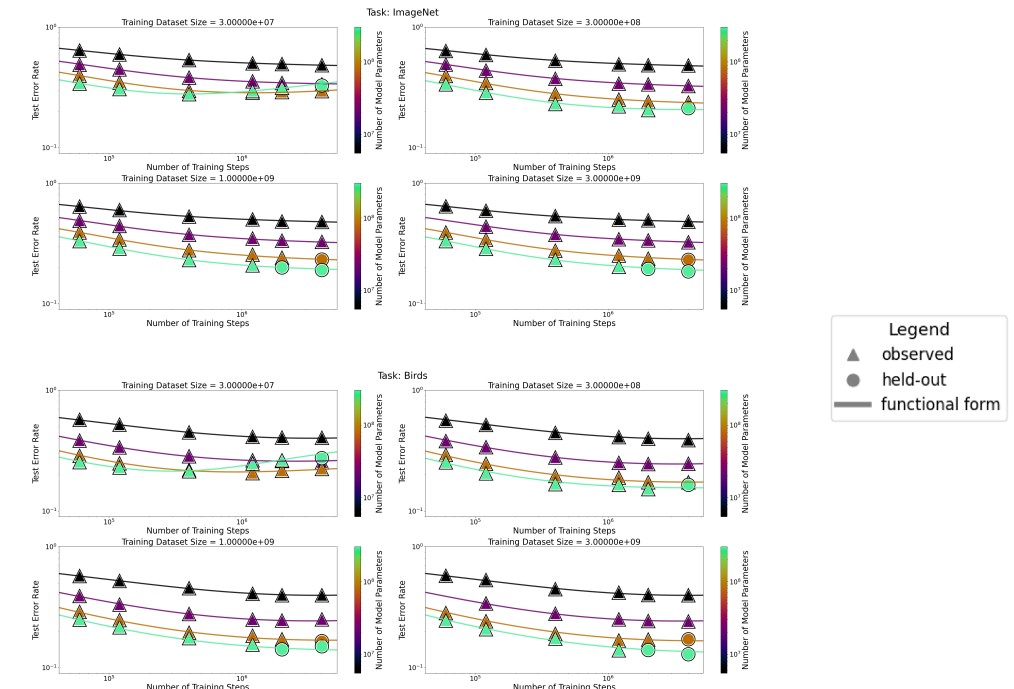

Figure 20: Extrapolation Results of A3 functional form on trivariate scaling behavior of downstream vision performance. See Section 4.1 for more details.

## 15.2 PLOTS OF LANGUAGE EXTRAPOLATION RESULTS

### 15.2.1 TRIVARIATE

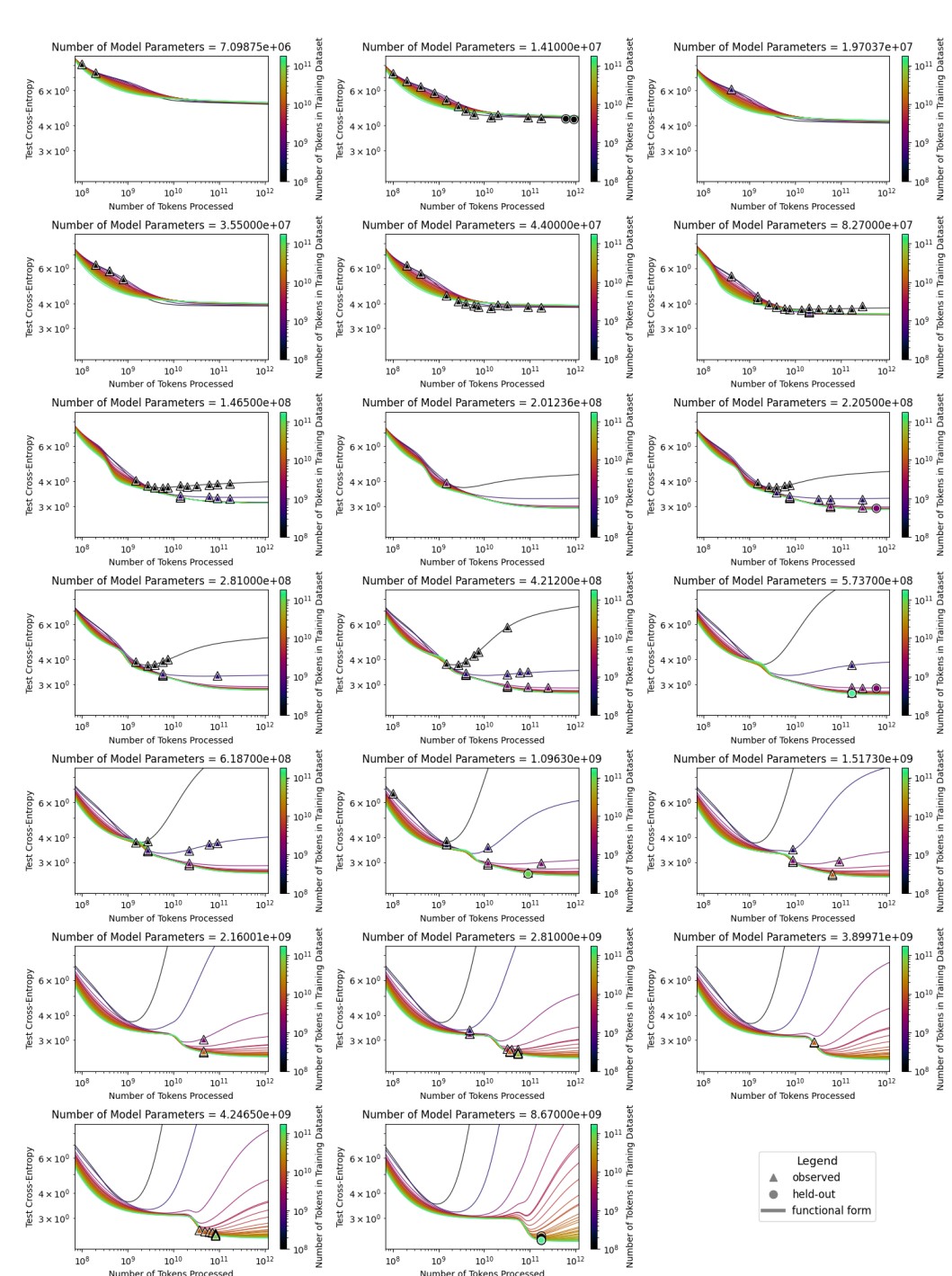

Figure 21: Extrapolation Results of UNSL on trivariate scaling behavior of language performance. All 20 plots are slices of single functional form fit to a single trivariate scaling behavior. The title of each plot represents the number of model parameters, the x-axis of each plot represents the number of training steps times the batch size, and the color bar of each plot represents the training dataset size. The See Section 4.2 for more details.

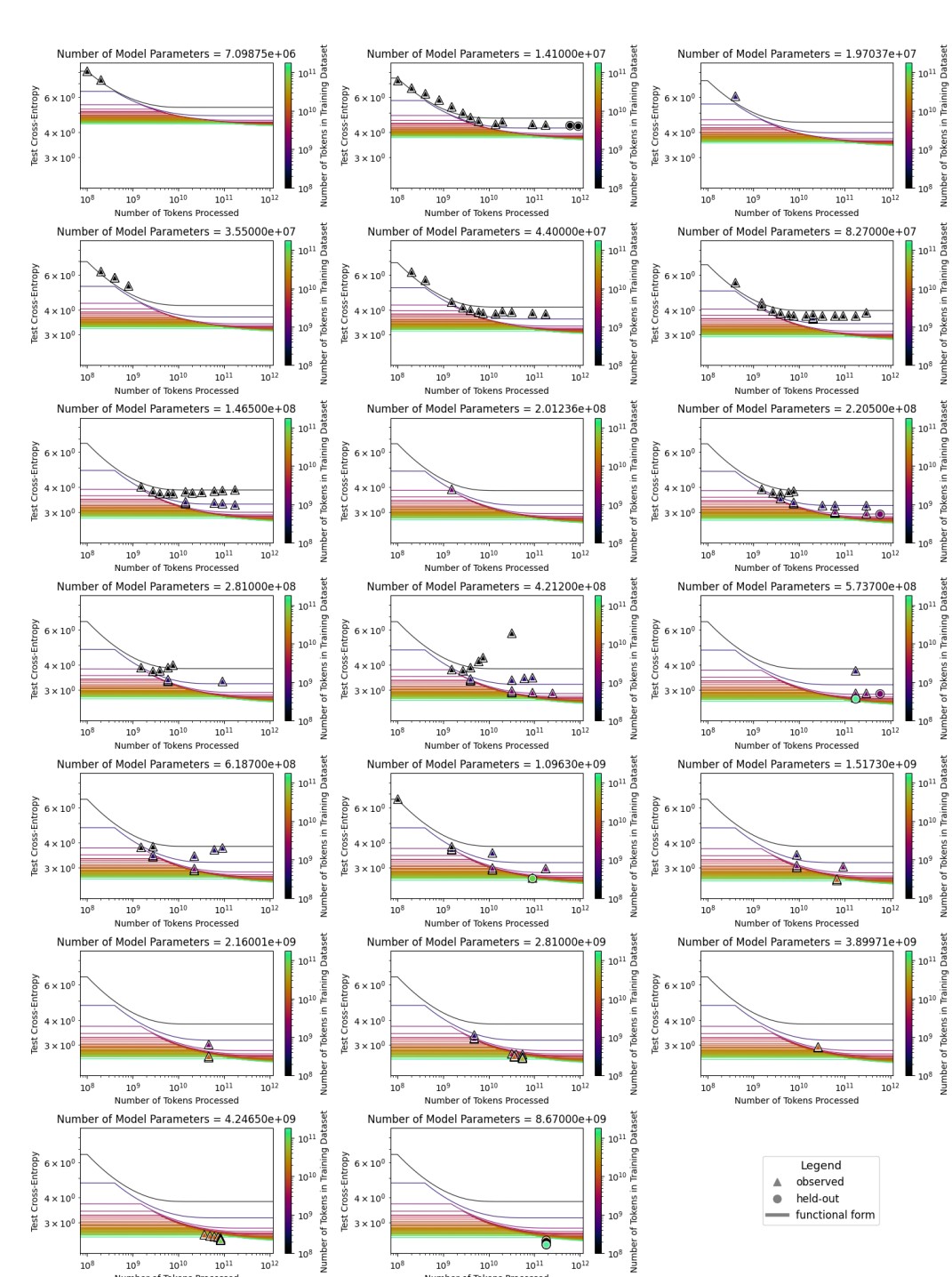

Figure 22: Extrapolation Results of "DC" functional form of Muennighoff et al. (2023) on trivariate scaling behavior of language performance. All 20 plots are slices of single functional form fit to a single trivariate scaling behavior. The title of each plot represents the number of model parameters, the x-axis of each plot represents the number of training steps times the batch size, and the color bar of each plot represents the training dataset size. The See Section 4.2 for more details.

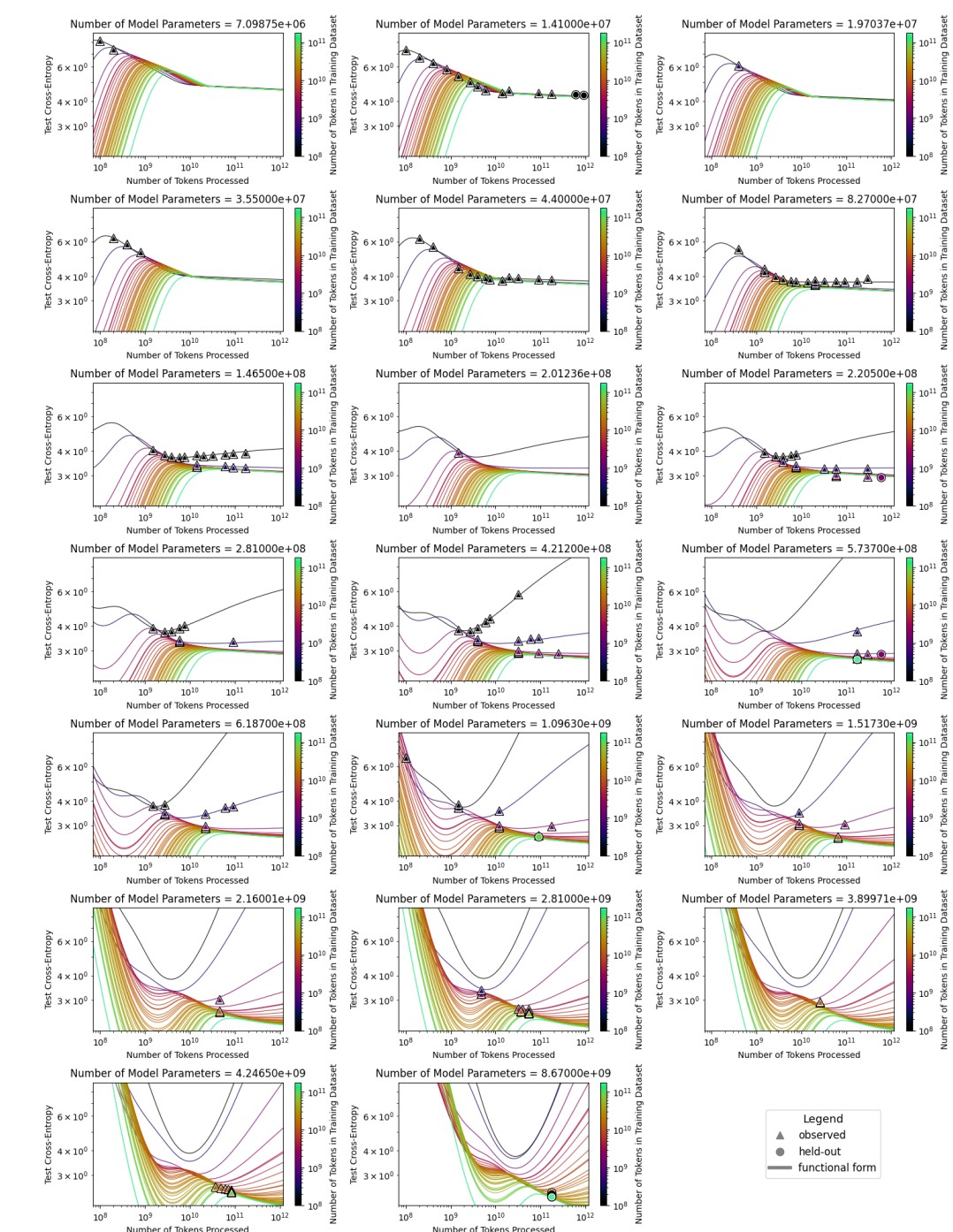

Figure 23: Extrapolation Results of A1 functional form on trivariate scaling behavior of language performance. All 20 plots are slices of single functional form fit to a single trivariate scaling behavior. The title of each plot represents the number of model parameters, the x-axis of each plot represents the number of training steps times the batch size, and the color bar of each plot represents the training dataset size. The See Section 4.2 for more details.

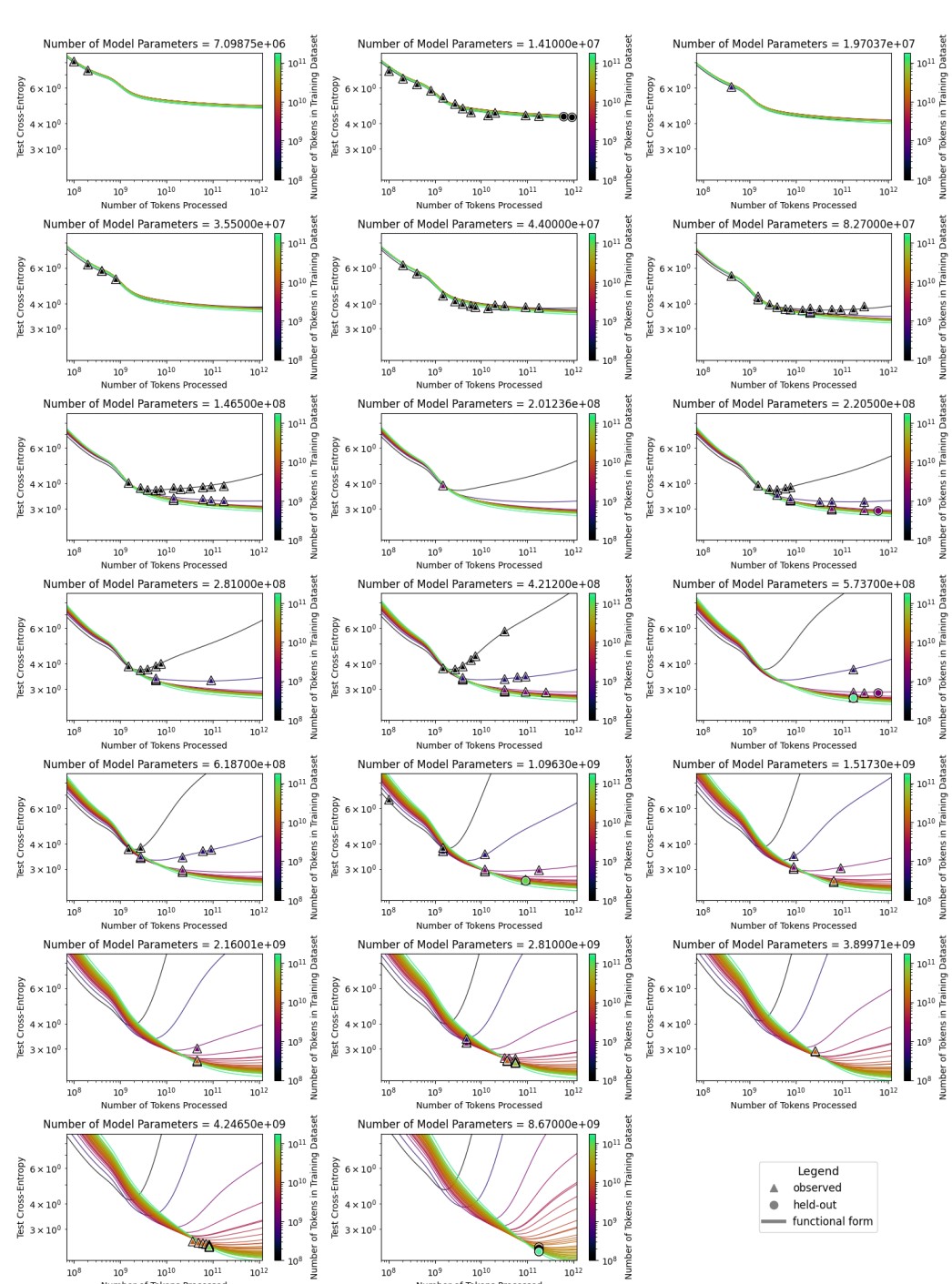

Figure 24: Extrapolation Results of A2 functional form on trivariate scaling behavior of language performance. All 20 plots are slices of single functional form fit to a single trivariate scaling behavior. The title of each plot represents the number of model parameters, the x-axis of each plot represents the number of training steps times the batch size, and the color bar of each plot represents the training dataset size. The See Section 4.2 for more details.

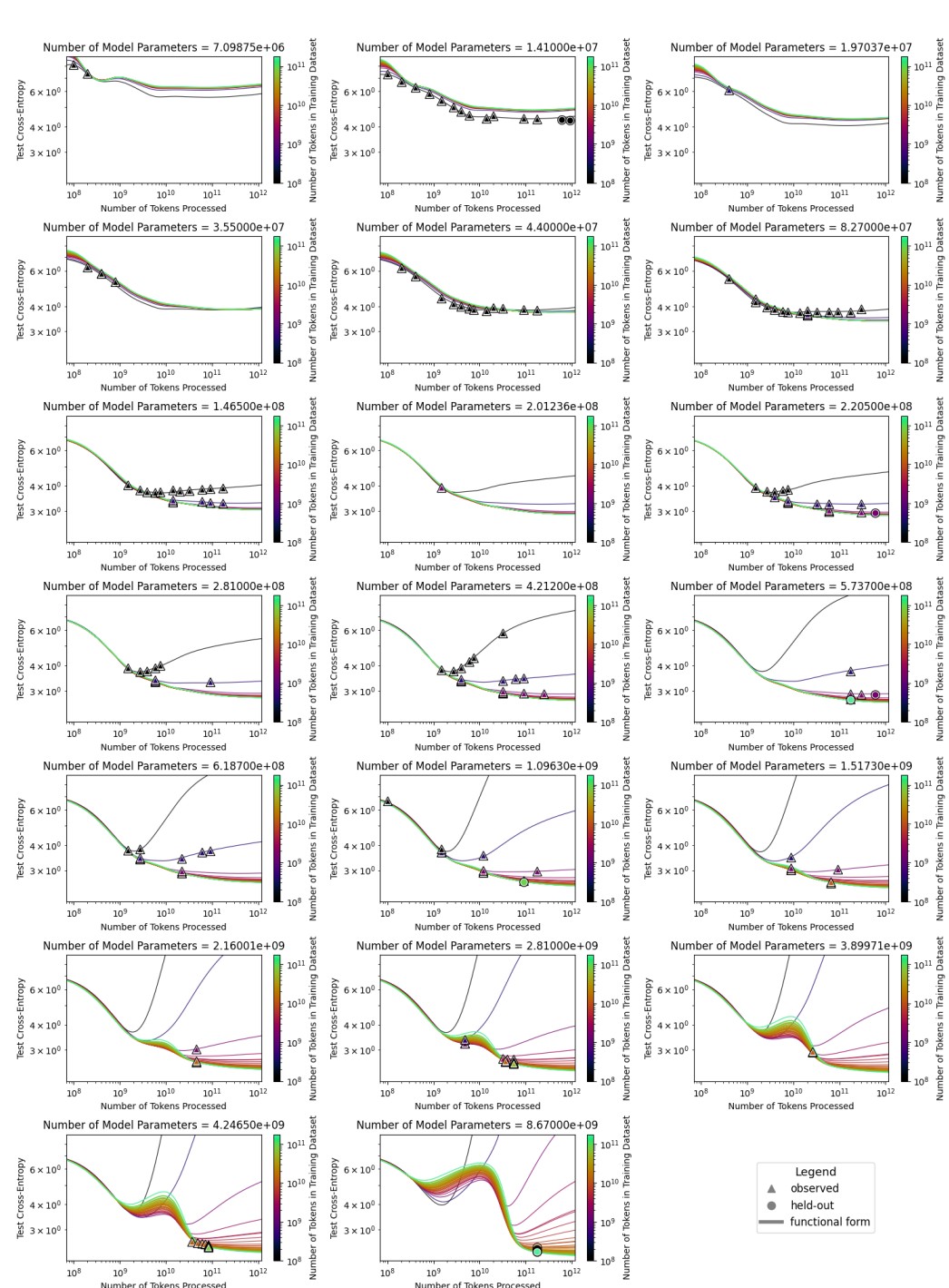

Figure 25: Extrapolation Results of A3 functional form on trivariate scaling behavior of language performance. All 20 plots are slices of single functional form fit to a single trivariate scaling behavior. The title of each plot represents the number of model parameters, the x-axis of each plot represents the number of training steps times the batch size, and the color bar of each plot represents the training dataset size. The See Section 4.2 for more details.

### 15.2.2 BIVARIATE

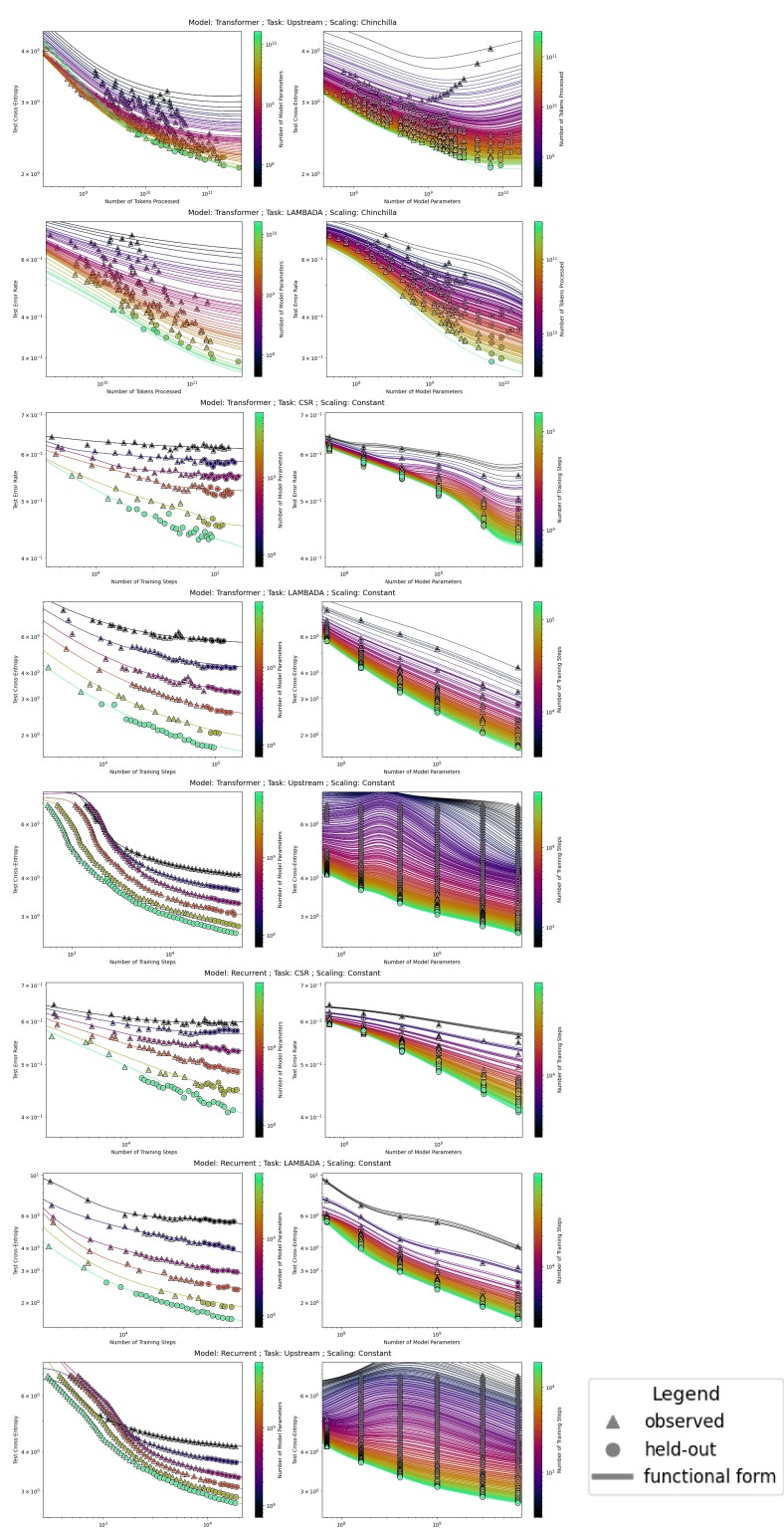

Figure 26: Extrapolation Results of UNSL on bivariate scaling behavior of downstream (and upstream) language performance. See Section 4.2 for more details.

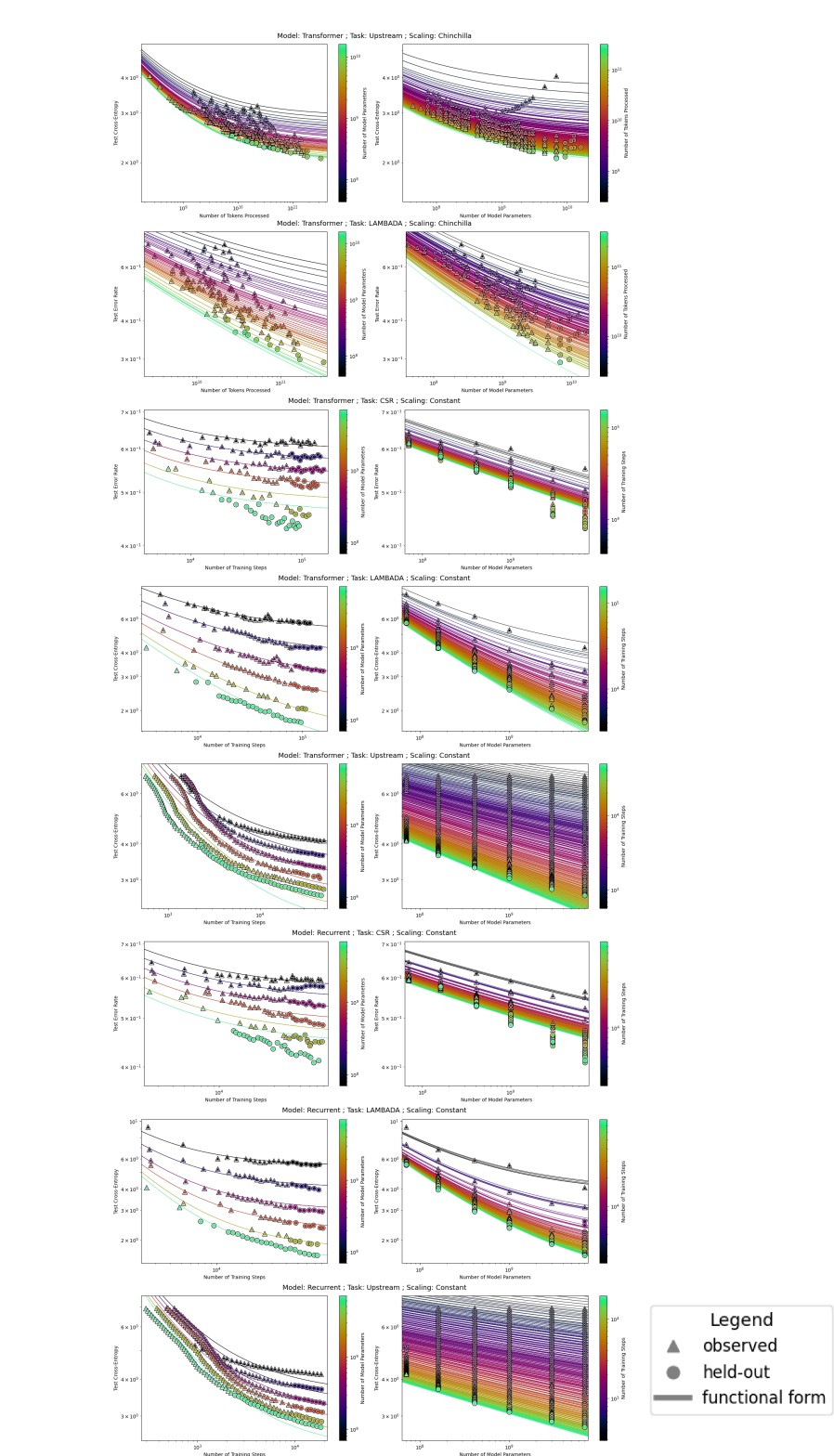

Figure 27: Extrapolation Results of "CF" functional form of Hoffmann et al. (2022) on bivariate scaling behavior of downstream (and upstream) language performance. See Section 4.2 for more details.

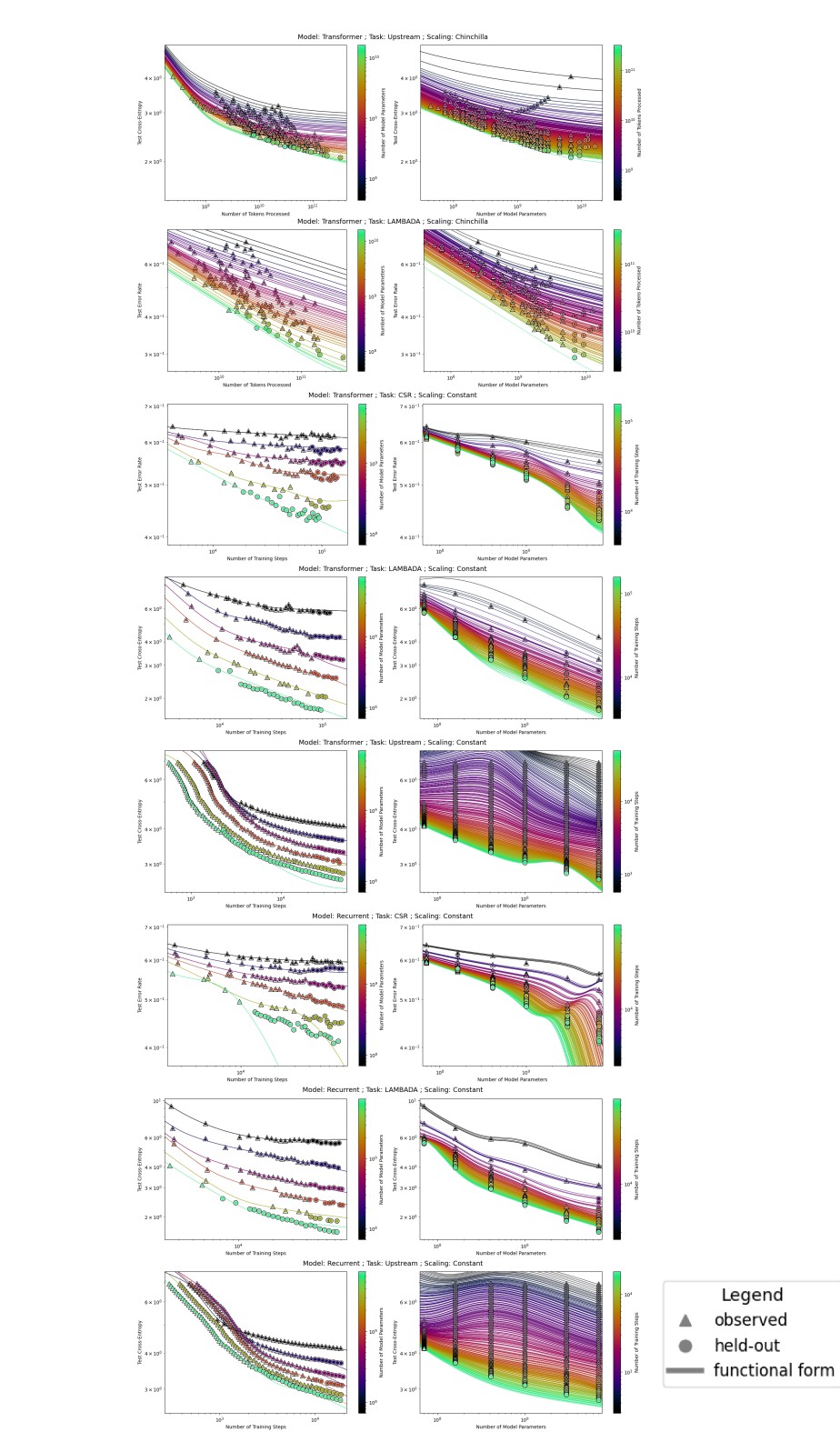

Figure 28: Extrapolation Results of A1 functional form on bivariate scaling behavior of downstream (and upstream) language performance. See Section 4.2 for more details.

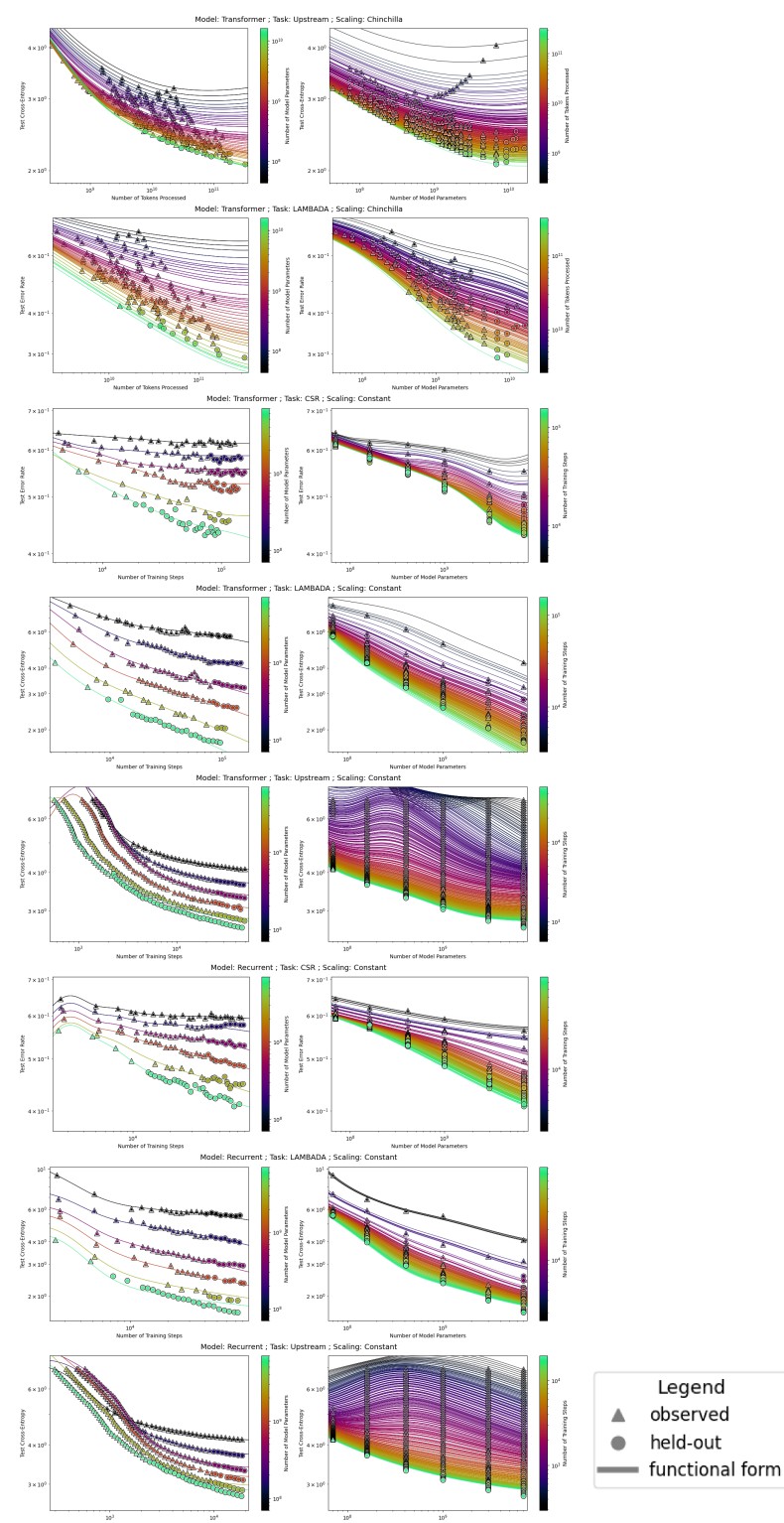

Figure 29: Extrapolation Results of A2 functional form on bivariate scaling behavior of downstream (and upstream) language performance. See Section 4.2 for more details.

## 16 UNSL ACCURATELY EXTRAPOLATING TO SCALES AN ORDER OF MAGNITUDE LARGER IN MULTIPLE DIMENSIONS SIMULTANEOUSLY

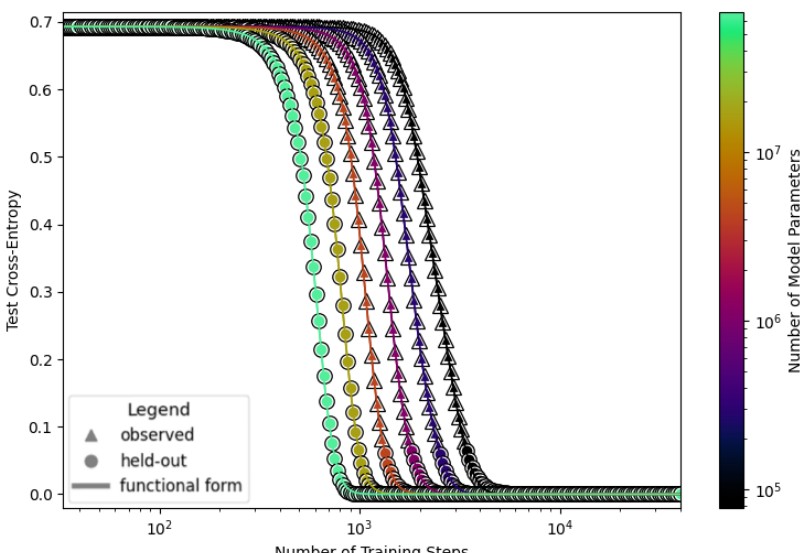

Figure 30: Extrapolation of UNSL on scaling behavior of an MLP trained for a single epoch on the (n, k)-sparse parity task (with $n = 40$ and $k = 4$) of Barak et al. (2022). Each point in the plot is the mean of greater than 100 seeds. See Section 16 and Appendix 11 for more details.

In Figure 30, UNSL accurately extrapolates to scales an order of magnitude larger in multiple dimensions simultaneously.

## 17 OBTAINING THE COMPUTE-OPTIMAL VALUES OF THE INPUT DIMENSIONS

Let $\mathcal{D}$ be the index set that contains the indexes of dimensions of $(x_1, ..., x_m)$ that directly contribute to amount of training compute used (e.g. number of model parameters, number of training steps, etc.). Let $\mathcal{H}$ be the index set that contains the indexes of dimensions of $(x_1, ..., x_m)$ that do not directly contribute to amount of training compute used (e.g. learning rate, standard deviation of weights at initialization, etc.). $C$ is amount of training compute used. $C_0$ is a constant (e.g. equal to 6 in Hoffmann et al. (2022)) such that $C_0 = C/(\prod_{i \in \mathcal{D}} x_i)$. $\lambda$ is a Lagrange multiplier.

To obtain the values of $(x_1, ..., x_m)$ that yield the lowest value of $y$ for a given value of $C$, one solves following system of equations:

$$\frac{\partial y}{\partial x_\ell} + \lambda \frac{C}{x_\ell} = 0, \ell \in \mathcal{D},$$

$$\frac{\partial y}{\partial x_v} = 0, v \in \mathcal{H},$$

$$C - C_0 \prod_{\ell \in \mathcal{D}} x_\ell = 0.$$

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
