# OpenReview forum: "Unified Neural Scaling Laws"
_ICLR.cc/2026/Conference — Submitted to ICLR 2026_

### Official Review · Reviewer_FXnf · 2025-10-17

**Soundness:** 2
**Presentation:** 2
**Contribution:** 4
**Rating:** 2
**Confidence:** 3

**Summary:**

The paper proposes a functional form for scaling laws across domains and architectures. The scaling law obeys a number of intuitive desiderata and generalizes a number of prior observed scaling behaviors. Empirically, the scaling law predicts performance very precisely across models and datasets.

**Strengths:**

The biggest strength in my view is the very strong predictive power of UNSL. The experiments in this paper are extensive and convincing. The authors don't compare with many baselines (primarily only DC/CF)- but frankly, this is fine given the very substantial improvement in predictive power that UNSL provides relative to the baselines here. It is hard to imagine a prior proposed scaling law with this strong fit to data. I overall feel that finding strong functional forms of scaling laws is a very important research direction, so I think this paper has the potential to be an excellent contribution to the field.

Also, the purple box in section 4 is a nice touch!

**Weaknesses:**

My main concern with this paper is that the proposed scaling law is incredibly expressive- in fact, it's so incredibly expressive that it's hard to imagine it couldn't fit the data. Its comparable to if performance were predicted by an expressive neural network that took in the $x_i$ as input.

Now, this would be ok if either 1) UNSL were theoretically motivated or 2) UNSL were shown to be the minimal scaling law explaining the empirical data. Regarding 1, UNSL obeys some nice desiderata but is not strictly theoretically motivated like some other common scaling laws in the literature. Perhaps the paper would be more convincing if the authors could prove that UNSL is the unique scaling law obeying desiderata 1-8. Regarding 2, the the authors do attempt to show that UNSL is minimal with the ablations A1-A3. Unfortunately, given the large number of moving parts in UNSL, I believe more ablations are necessary to justify each and every component of UNSL. It's hard for me to imagine that there isn't another scaling law with similar complexity to UNSL that also achieves strong predictive performance.

Minor comments
- Please avoid vertical lines in tables (use booktabs style)
- Typically, the appendix is placed after references
- The text in some of the figures is so small that it is unreadable (e.g. Figure 10)

**Questions:**

- Can UNSL be proven to be the unique scaling law obeying the authors' desiderata?
- Can the authors provide additional ablations beyond A1-A3?
- See minor comments above

---

> ### Author Response · Authors · 2025-11-22
> **Reply to reviewer FXnf**
>
> We thank the reviewer for useful feedback and attempt to clarify below all the concerns raised.
>
> > 1. UNSL obeys some nice desiderata but is not strictly theoretically motivated like some other common scaling laws in the literature.
>
> The common scaling laws (of large deep networks trained on natural data) in the literature are not “strictly theoretically motivated”. They are phenomenological fits/extrapolations, with theory coming afterward (in follow-up papers) in simplified toy settings.
>
> Theoretical derivation of the scaling functional form from the first principles is clearly hard; i.e., there currently does not exist a widely accepted, empirically accurate mechanistic theory that explains why scaling behaviors of large deep neural networks trained on natural data often follow/involve power laws in such a wide variety of settings, despite power law scaling behavior of multilayer neural networks first being observed over 30 years ago in https://papers.nips.cc/paper/1993/file/1aa48fc4880bb0c9b8a3bf979d3b917e-Paper.pdf . Note that the field of neural scaling laws is inherently empirical by its nature, precisely because performing (useful) theoretical analysis of very large, deep, and complex networks’ performance (and other behaviors) becomes exceedingly (some would say prohibitively) complicated. Thus, the empirical science approach to neural performance is necessary, akin to studies of complex network behavior in other fields such as statistical physics (e.g. see 1:16:39 onward of https://youtu.be/CR45mBkSH7g?t=4599)
>
> > 2. Can the authors provide additional ablations beyond A1-A3?
>
> There does not exist any paper in the neural scaling laws literature that compares the performance of more functional forms (e.g. ablation functional forms) than our paper does; as a result, the request to provide additional ablations beyond A1-A3 is out of scope.
>
> > 3. Can UNSL be proven to be the unique/minimal scaling law obeying the authors' desiderata and/or explaining the empirical data?
>
> There does not exist any paper in the neural scaling laws literature that proves that the functional form of a paper is the unique/minimal scaling law obeying a paper’s authors' desiderata or explaining the empirical data; as a result, the request to “prove that the functional form of the paper is the unique/minimal scaling law obeying the authors' desiderata or explaining the empirical data” is out of scope.

---

> > ### Comment · Reviewer_FXnf · 2025-11-22
> > **Thank you for your reply**
> >
> > I agree with the authors that purely empirical studies of scaling are critical to pushing the field; not all scaling studies need to be theoretically motivated. However, the key to these empirical studies is *abstraction*: they aim to encapsulate complex phenomena with simple, interpretable laws. Jared Kaplan's talk cited by the authors is a good example of this: the scaling laws he proposes are easy-to-interpret, but imperfect abstractions of the real phenomenon. This is not merely because it was a talk: the Kaplan et al., 2020 paper proposes relatively simple power laws despite relatively extensive empirical analysis.
> >
> > Unfortunately, as some of the other reviewers mentioned, I'm not convinced UNSL is sufficiently interpretable: as the other reviewers note it is "extremely intricate" and "feels more like a complex black box." In the absence of this interpretability, I believe some notion of UNSL being unique/minimal or additional ablations are necessary.
> >
> > The authors raise the point that prior papers on scaling laws do not consider ablations of their functional forms or prove their minimality. While I appreciate this appeal to precedent, I also don't believe prior scaling laws are as complex and uninterpretable as UNSL is. Prior scaling laws don't need to show functional form ablations or minimality because they are sufficiently interpretable to begin with.

---

### Official Review · Reviewer_e5yM · 2025-10-30

**Soundness:** 3
**Presentation:** 3
**Contribution:** 2
**Rating:** 4
**Confidence:** 3

**Summary:**

This paper proposes a new neural scaling law formula which can be fit to empirical data. This formula contains multivariate broken neural scaling laws as a special case but also incoporates overfitting effects for multi-epoch training. They show that their fits can extrapolate accurately on computer vision, language and sparse parity tasks.

**Strengths:**

I think this paper attempts to provide a more general and flexible framework for empirical neural scaling law fits. The authors also included a heroic number of experiments attempting to characterize the quality of their model's extrapolation.

**Weaknesses:**

**Theoretical Motivation** Some of the new terms that the authors introduce are not immediately intuitive, especially the "oppositional force from hyperparameters." Is there a minimal toy model that motivates these terms? In general, I would appreciate any minimal models that motivate inclusion of each term (for instance it appears double descent motivates the broken scaling law in data + model params if I understand correctly?).

**More complex can be less interpretable** While this model has more predictive power, I am concerned that we could risk developing increasingly complex functions to approximate the scaling laws which makes design decisions (eg compute optimal joint scaling of train time and params) more complex and less interpretable. Chinchilla scaling laws are nice in this regard. As a possibly unfair reductio, at what point do we use a non-parametric or deep network to fit the scaling law itself?

That said, I do appreciate any improvements in predictive accuracy. I would also be especially interested if the **gaps in explainability** between different scaling law models can be attributed to phenomena that we can invest time understanding scientifically. This is why I am curious about minimal models that capture failure modes of classic scaling laws.

**Questions:**

1. How do you decide how many breakpoints to include in the fit? What would happen if the breakpoint count were much larger than the true number of breaks?
2. Can you easily deduce compute optimal scaling strategies ($\min_{N,T} \mathcal L(T,N) \text{s.t.} TN = C$) from this unified scaling law?

---

> ### Author Response · Authors · 2025-11-22
> **Reply to reviewer e5yM**
>
> We thank the reviewer for the comments and provide a detailed response that addresses the reviewer’s concerns.
>
> > 1. Theoretical Motivation. Some of the new terms that the authors introduce are not immediately intuitive, especially the “oppositional force from hyperparameters.” Is there a minimal toy model that motivates these terms? In general, I would appreciate any minimal models that motivate inclusion of each term (for instance it appears double descent motivates the broken scaling law in data + model params if I understand correctly?).
>
> Minimal toy models are accurately fit to (and accurately extrapolate) real scaling behavior data (that each correspond to individual desiderata) in Appendix 14; note that “Appendix 14.7 and 14.8” (and the aspects related to Figure 7 in Appendix 14.5) specifically correspond to “oppositional force from hyperparameters.”  In Appendix 13, UNSL is shown to satisfy the same desiderata that the minimal toy models show (in Appendix 14) to be empirically true; note that “Appendix 13.7 and 13.8” (and the aspects related to Figure 7 in Appendix 13.5) specifically correspond to “oppositional force from hyperparameters.”
>
> In addition, we include 3 ablation functional forms (called A1, A2, and A3) evaluated on all scaling behaviors of Section 4 to show that functional forms that are more minimal (with regards to the number of additive symmetries that are incorporated) than UNSL yield worse extrapolations than UNSL.
>
> > 2. How do you decide how many breakpoints to include in the fit?
>
> As explained in Appendix 10:
>
> The values of n (i.e. the number of breaks) that yield the lowest extrapolation error can be obtained as follows. Split the set of observed points (i.e. the triangle-shaped points, NOT the circle-shaped points) used for fitting into two sets, a validation set and a training set; for each (i.e. every) point in the validation set, the training set should not contain a point that is simultaneously larger than each (i.e. every) $x$ dimension $(x_1, …, x_m)$ of that validation set point. The values of $n$ with the lowest validation error when fitting on the remaining training points are then used; (and the hold-out points (i.e. the circle-shaped points) are still held out to evaluate extrapolation RMSLE).
>
> > 3. What would happen if the breakpoint count were much larger than the true number of breaks?
>
> If $n$ (i.e. the number of breaks) is too small, one underfits. If $n$ is too large (and the number of points for fitting is too small), one overfits. Note that in Appendix 12 “Effect of Varying the Number of Observed Points Used for Fitting UNSL Functional Form”, we observe that using 9 observed points is sufficient to prevent overfitting.
>
> > 4. I would also be especially interested if the gaps in explainability between different scaling law models can be attributed to phenomena that we can invest time understanding scientifically.
>
> Yes, phenomena that we can invest time understanding scientifically are the "oppositional force from hyperparameters" and the "oppositional force from overfitting".
>
> > 5. Can you easily deduce compute optimal scaling strategies from this unified scaling law?
>
> Yes, see Appendix 17 titled “Obtaining the Compute-Optimal Values of the Input Dimensions”.

---

> > ### Comment · Reviewer_e5yM · 2025-11-24
> > **response to rebuttal**
> >
> > I thank the authors for their responses. I will maintain my current score.

---

### Official Review · Reviewer_WM28 · 2025-10-31

**Soundness:** 3
**Presentation:** 2
**Contribution:** 2
**Rating:** 4
**Confidence:** 4

**Summary:**

This paper introduces a highly general functional form meant to describe how neural network performance scales jointly with model size, dataset size, training steps, and hyperparameters. It aims to go beyond the standard two-variable scaling laws (like Kaplan or Chinchilla) toward a unified predictive framework.
Overall Impression
The work is technically ambitious and very thorough, but the motivation and clarity of purpose could be stronger. I appreciate the attempt to unify the many partial scaling laws into one consistent form, but at first read it’s not clear why this particular functional structure is the right way to do it. The introduction lays out the goal of a “unified law,” but the number of nested definitions and the complexity of Equations (1–4) make it hard to see the intuition behind them or to grasp what each part captures in terms of actual network behavior.
As one colleague put it after a quick look: “I couldn’t find the smoking-gun. We already have scaling laws that extrapolate accurately over 10⁴× changes in compute — so what exactly does this improve?” I share that reaction to some extent. The authors demonstrate clear empirical accuracy, but it’s not obvious whether UNSL provides qualitatively new predictive power or simply adds flexibility to fit more phenomena (like overfitting and poor hyperparameter choices) that existing forms treat as noise.

**Strengths:**

The authors collect an unusually broad suite of benchmarks: multiple architectures (ViT, BiT, Mixer, Transformer), both vision and language domains, and even a sparse-parity setup that tests generalization. That level of coverage is commendable.  Across these tasks, UNSL tends to outperform simpler baselines (CF, DC, A1–A3) in RMSLE, often by a wide margin. The experimental design is careful, and the visualizations of extrapolation (e.g., Fig. 2) are clear and fair. The paper explicitly models phenomena like overfitting and learning-rate effects, which most scaling laws ignore. That’s a valuable step toward realism.

**Weaknesses:**

Clarity and motivation. The mathematical form is extremely intricate — nested sums, products, and “oppositional forces.” Even with the appendices, it’s hard to understand what behavior each term is meant to represent. A few schematic diagrams or concrete toy examples would help a lot. Predictive novelty. It’s not clear whether UNSL truly extends the predictive range beyond existing laws. The extrapolation plots look accurate, but they don’t seem to cover very wide ranges (e.g., the 10^4-fold compute spans shown in Kaplan et al. 2020 or Hoffmann et al. 2022). Demonstrating a case where UNSL predicts something current methods fail to capture would strengthen the contribution. Interpretability vs. overfitting. Because the form includes many adjustable constants, it’s possible that it simply absorbs “parasitic” phenomena like sub-optimal hyperparameters rather than isolating fundamental scaling behavior. The fitting process (choosing n, S, λ, and seed selection) adds further degrees of freedom. Some sensitivity analysis or cross-validation results would help rule out over-fitting to small datasets. Relation to existing scaling theories. There’s an active line of work tying scaling exponents to the intrinsic dimensionality of the data manifold. It would be useful to know whether UNSL is compatible with or contradicts those ideas. Right now, the connection is left unexplored.

**Questions:**

The paper would benefit from showing some practical payoff. For example: can this form be used to predict optimal hyperparameters (learning rate, batch size) that minimize loss at a given compute budget? Or to forecast the point of diminishing returns? Without such demonstrations, it risks being an elegant but abstract curve-fitting tool.  The breakpoints (“hyperbreaks”) look numerous and somewhat local; it’s not clear whether UNSL can extrapolate far beyond observed data or whether each region must be densely sampled. The claim that one must fit “sufficiently close” to each hyperbreak raises concern about general utility. The authors position UNSL as an extension of “broken” scaling laws (Caballero et al., 2023), and that’s reasonable. But some comparison with μP (maximal update parameterization) and CompleteP frameworks would be useful. Those approaches explicitly normalize hyperparameters across scale, effectively removing one source of non-scaling behavior. If UNSL primarily captures those sub-optimal regimes, then its “extra accuracy” may just reflect modeling of effects that better parameterizations already avoid.

---

> ### Author Response · Authors · 2025-11-21
> **Reply to reviewer WM28**
>
> We thank the reviewer for the comments and provide a detailed response that addresses the reviewer’s concerns.
>
> > 1. over-fitting to small datasets
>
> In Appendix 12 titled “Effect of Varying the Number of Observed Points Used for Fitting UNSL Functional Form”, we vary the number of observed points used for fitting UNSL functional form from 9e0 to 9e2, and observe that UNSL accurately extrapolates scaling behavior when only a small number of observed points (i.e. 9 points) are used for fitting UNSL functional form. This is evidence that UNSL is not over-fitting to small datasets.
>
> > 2. A few concrete toy examples would help.
>
> Concrete minimal toy models are accurately fit to (and accurately extrapolate) real scaling behavior data (that each correspond to individual desiderata) in Appendix 14. In Appendix 13, UNSL is shown to satisfy the same desiderata that the concrete minimal toy models show (in Appendix 14) to be empirically true.
>
> > 3. Predictive novelty. It’s not clear whether UNSL truly extends the predictive range beyond existing laws. The extrapolation plots look accurate, but they don’t seem to cover very wide ranges (e.g., the 10^4-fold compute spans shown in Kaplan et al. 2020 or Hoffmann et al. 2022). Demonstrating a case where UNSL predicts something current methods fail to capture would strengthen the contribution.
>
> Section 4 and Appendix 15 contain dozens of cases in which “UNSL predicts something current methods fail to capture”: Every case in which UNSL yields a lower extrapolation error than CF (CF is the functional form from Kaplan et al. 2020 and Hoffmann et al. 2022) and DC (which is an extension of CF from Muennighoff et al. 2023 that becomes equivalent to CF when training dataset size is so large that one only trains for one epoch) is a case in which “UNSL predicts something current methods fail to capture”.
>
> We use the exact “10^4-fold compute span” scaling behavior data from Hoffmann et al. 2022 (obtained via correspondence with Hoffmann et al. 2022) in the plots in Appendix 15.2.2 (titled “Bivariate”) with the word “Chinchilla” in the plot title and in the rows of Table 4 with the word “Chinchilla” in the “Scaling” column. As can be seen in Table 4, UNSL extrapolates more accurately than CF (i.e. the functional form from Kaplan et al. 2020 and Hoffmann et al. 2022) on this scaling behavior data.
>
>
> > 4. can UNSL extrapolate far beyond observed data?
>
> In Appendix 16, we show that UNSL accurately extrapolates to scales an order of magnitude larger (than the maximum (along each of multiple input axes) of the observed data) in each of multiple dimensions simultaneously.
>
> > 5. Relation to existing scaling theories. There’s an active line of work tying scaling exponents to the intrinsic dimensionality of the data manifold. It would be useful to know whether UNSL is compatible with or contradicts those ideas. Right now, the connection is left unexplored.
>
> Recall that papers such as Caballero et al. (2023) have observed that univariate scaling behaviors follow a smoothly broken power law functional form; i.e., they have observed that within a univariate scaling behavior the value of the exponent of that power law relationship can change to a different exponent as the individual $x$ (i.e. input) dimension (of that univariate scaling behavior) varies. Other than that fact (which UNSL enforces via Desideratum 1) that the value of the exponent of that power law relationship can change to a different exponent as the individual $x$ (i.e. input) dimension (of a univariate scaling behavior) varies, UNSL does not contradict current versions of the line of work tying scaling exponents to the intrinsic dimensionality of the data manifold.
>
> > 6. can this form be used to obtain optimal hyperparameters (learning rate, batch size) that minimize loss at a given compute budget?
>
> Yes, see Appendix 17 titled “Obtaining the Compute-Optimal Values of the Input Dimensions”.
>
> > 7. μP (maximal update parameterization) and CompleteP frameworks
>
> The oppositional force of hyperparameters still exists when using CompleteP and μP. CompleteP and μP present a way to obtain the optimal hyperparameters at various scales, i.e. similar to the goal of Appendix 17 titled “Obtaining the Compute-Optimal Values of the Input Dimensions” of our UNSL paper.

---

### Official Review · Reviewer_g3Rd · 2025-11-01

**Soundness:** 2
**Presentation:** 2
**Contribution:** 2
**Rating:** 2
**Confidence:** 4

**Summary:**

This paper proposes Unified Neural Scaling Laws (UNSL) — a general composable functional form that provides a lot of flexibility to which kinds of empirical relations between predictors (model size, training steps, ...) and observables (loss, downstream evaluations, ...) can be fitted with UNSL. UNSL allows for capturing a wide array of behaviors: breaks between different power-law trends (as in broken neural scaling laws); various bottlenecks; "oppositional force" of overfitting or suboptimal choice of hyperparameters; misperformance limits.

The authors define eight desiderata (Section 2.2) to set the desired properties of scaling laws. Then, they demonstrate in Appendix 13, with varying degree of rigor and detail (Appendix 13), that UNSL satisfies all 8 disiderata. Lastly, in Appendix 14, the authors provide a brief empirical evidence for each disiderata.

In the experimental part, the authors take various open datasets of (predictors, evaluation metric) pairs from previous scaling studies and fit them with UNSL, obtaining lower extrapolation error. These open sets are complemented by additional experiment with learning sparse parity task with MLP. The UNSL fitting process is described in Appendix 10, and amounts to 20000 steps and 20 random seeds of KFAC-JAX second order optimization method to be run on a set of UNSL hyperparameters (such as the number of breaks) to determine the best ones for the final fit.

**Strengths:**

The authors pursue an ambitious goal of unifying various scaling behaviours in a single functional form. To this end, they provide a suitable high-level structure:
- formulate a set of desired properties or scaling behaviors (through 8 desiderata points)
- empirically shows the need for these properties
- connect the set of properties to the scaling law functional law (through showing that UNSL satisfies 8 desiderata)

**Weaknesses:**

My main concern is the excessive structural complexity of the proposed UNSL. This has several downsides
- *Paper focus.* Almost a third of the paper is taken up just by writing down and providing comments on UNSL form. As a result, many conceptually important steps, like empirical justification of 8 proposed disiderata points, and connecting them to UNSL are fully moved to the appendix. Likewise, experimental validation of the utility of UNSL takes a lot of space just to plainly state the settings and results. Even then, the whole scope of UNSL feels barely covered by the presented experiments.
- *Analysis depth*. Most of the aspects feel to be discussed/analyzed very briefly (consequence of UNSL complexity and scope), making it hard to convince a reader of the paper's claims.
- *Switch from modeling actual model behavior to brute force fitting with an expressive functional form.* As the authors rightly mention in the introduction, adding more predicting factors or fitting parameters reduces the conditional entropy of evaluation metrics to be fitted by a scaling law. Naturally, UNSL with many hyperparameters fits the data very well due to lots of parameters (vs <5 for classical chinchilla scaling law). Then, it is unclear whether the good fit comes from the model in the experiment really exhibiting the 8 disiderata behaviors encoded in UNSL, or from the sheer expressive power of UNSL with many parameters.
- *Practical usefulness.* An important aspect of classical scaling laws is simplicity. This makes them robust and easily verifiable (e.g. a single power-law can be visually verified by plotting data in log scale). UNSL feels more like a complex black box fitted with a second-order optimization method. Then it is very hard to get an intuition about the behaviour of the fitted model; draw conclusions, like compute optimal allocations; and scary to use UNSL for extrapolation to expensive large-scale regimes.

At least from my perspective, focusing on a few novel aspects of UNSL (i.e. the respective desiderata points) and showing that they are essential for explaining the behaviour of a certain set of deep learning systems would be much more interesting and useful work.

Also, the writing of the manuscript can be greatly improved - currently it reads as something between a paper and a working journal/draft

**Questions:**

- Can disiderata points be more rigorously formalized, so that satisfaction of these points by UNSL could be formulated as a theorem?
- Are there some constraints on UNSL parameters? For example, it is reasonable to assume that $a_i$ are positive, but it is not mentioned in the text.
- What are the principles/practical guidelines behind choosing the sets of predictors $U_r$? The straightforward search over all subsets of $\{1,\ldots,m\}$ might be hard due to the combinatorial explosion of all the possibilities.
- In the review, I was assuming that all the experiments, except for learning sparse parity task, are taken from prior work and open sources. In some cases, this is explicitly mentioned in the paper, but for other cases, I could not find an explicit comment. Please point out if I mistakenly confuse some of the original experiments of this work as borrowed from the literature.

---

> ### Author Response · Authors · 2025-11-21
> **Reply to reviewer g3Rd**
>
> We thank the reviewer for useful feedback and attempt to clarify below all the concerns raised.
>
> > 1. In the review, I was assuming that all the experiments, except for learning sparse parity task, are taken from prior work and open sources. In some cases, this is explicitly mentioned in the paper, but for other cases, I could not find an explicit comment. Please point out if I mistakenly confuse some of the original experiments of this work as borrowed from the literature.
>
> All experiments for which the experimental details are described in Appendix 11 are original experiments; all other experimental scaling behavior is from prior work.
>
> > 2. It is unclear whether the good fit comes from the model in the experiment really exhibiting the 8 desiderata behaviors encoded in UNSL, or from the sheer expressive power of UNSL with many parameters.
>
> Recall the following points:
> - a) Ablation A1 enforces Desiderata 1 and 2. Ablation A2 enforces Desiderata 1, 2, 3, and 4. Ablation A3 enforces Desiderata 1, 2, 3, 4, 5, 6, and 7. UNSL enforces all 8 desiderata.
> - b) Desiderata 3, 4, 5, 6, 7, and 8 are all just variants of the additive relationship (i.e. the summation) that occurs in Equation 5.
> - c) Equation 6 is simply an n=1 version of Ablation A1.
> - d) The expressivity of Equation 6 becomes equivalent to the expressivity of Equation 5 when parameter f from Equation 6 is equal to -1.
> - e) Due to points a, b, c, and d being simultaneously true, the following hypothesis classes each have the same supremal expressivity: A1, A2, A3, UNSL.
> - f) The extrapolation error (e.g. as shown by Table 1 of our paper) is ordered as follows: UNSL < A3 < A2 < A1.
> - g) Due to point a, “the set of desiderata enforced by UNSL” ⊃ “the set of desiderata enforced by A3” ⊃ “the set of desiderata enforced by A2” ⊃ “the set of desiderata enforced by A1”.
>
> Due to points e, f, and g being simultaneously true, it is clear that the good extrapolation (relative to A1, A2, and A3) of UNSL comes from the 8 desiderata behaviors encoded in UNSL, not from the sheer expressive power of UNSL.
>
> > 3. Experimental validation of the utility of UNSL takes a lot of space just to plainly state the settings and results.
>
> The “experimental validation of the utility of UNSL” (i.e. Section 4) is 3.5 pages, which is the same length as other papers that introduce a new functional form (e.g. Section 5 of https://proceedings.neurips.cc/paper_files/paper/2022/file/8c22e5e918198702765ecff4b20d0a90-Supplemental-Conference.pdf).
>
>
> > 4. Are there some constraints on UNSL parameters? For example, it is reasonable to assume that $a_i$ are positive, but it is not mentioned in the text.
>
> When the performance evaluation metric ($y$) is a metric for which lower is considered better (such as test error rate or test cross-entropy), all parameters that are not exponents are non-negative.
>
> > 5. UNSL feels to be discussed/analyzed very briefly.
>
> We spend 10 pages discussing/analyzing UNSL (3.5 pages in Section 2, 3 pages in Appendix 13, 3.5 pages in Appendix 14).
>
> > 6. What are the principles/practical guidelines behind choosing the sets of predictors $U_r$? The straightforward search over all subsets of $\\{1, … , m\\}$ might be hard due to the combinatorial explosion of all the possibilities.
>
> $U_r$ is the set of predictors that are available (i.e. that vary) in the experimental scaling behavior data. $\\{1, … , m\\}$ is the set of all predictors. For example, if training_dataset_size and number_of_training_steps are the only predictors that vary in the experimental scaling behavior data, then $U_r$ = {training_dataset_size, number_of_training_steps} even though there are additional predictors (e.g. number of model parameters) in the set $\\{1, … , m\\}$ that do not vary in the experimental scaling behavior data. The UNSL functional form is set up such that predictors that do not vary in the experimental scaling behavior data are implicitly folded into the values of the parameters $a_0$, $b_k$, and $d_{j_k}$.
>
> > 7. how to draw conclusions, like compute optimal allocations?
>
> See Appendix 17 titled “Obtaining the Compute-Optimal Values of the Input Dimensions”.
>
> > 8. Is it practically useful to use UNSL for extrapolation to expensive large-scale regimes?
>
> Section 4 and Appendix 15 contain dozens of cases in which UNSL accurately extrapolates to expensive large-scale regimes.

---

### Public Comment · ~Ethan_Caballero1 · 2026-06-18
**See arXiv version for longer version of this "Unified Neural Scaling Laws" paper with more results: https://arxiv.org/abs/2605.26248**

See arXiv version for longer version of this "Unified Neural Scaling Laws" paper with more results:

https://arxiv.org/abs/2605.26248

https://arxiv.org/pdf/2605.26248

---

### Meta-Review · Area_Chair_1Uib · 2026-01-07

**Summary:**

The authors study a method of having a highly unified form of scaling laws, which covers many different terms. The authors also found the scaling laws to be very predictive empirically with good extrapolation error.

However, the reviews overall responded negatively, with the following serious concerns:
- Excessive structural complexity, also including definitions and notations
- Weak intuition and interpretability
- Expressivity and overfit concerns
- Many degrees of freedom in fitting procedure
- Unclear qualitative novelty
- Lacking theoretical ground, and questionable if this is the minimal form

Overall, I tend to be somewhat sympathetic to the reviewers. From what I understand, the number of parameters in the scaling law form proposed by the authors can grow quite quickly with the number of predictors, hyperbreaks, misperformance limits, bottleneck terms etc. and can quickly grow to be overparameterized. This level of complexity should allow the function to fit most data points on a bounded domain, which is essentially what the reviewers were concerned about as well. The main issue as the reviewers also point out is that it's no longer clear if this complex form is useful and interpretable.

Considering that these concerns above were not particularly well addressed, and the overall sentiment being quite negative, I would recommend reject.

**Reviewer Concerns:**

- Excessive structural complexity, also including definitions and notations: partly addressed, but not simplified
- Weak intuition and interpretability: partly addressed, still lacking interpretability
- Expressivity and overfit concerns: partly addressed with empirical check, but skepticism remain
- Many degrees of freedom in fitting procedure: partly addressed, but still require substantial hyperparameter search/dof
- Unclear qualitative novelty: partly addressed, although not agreed upon
- Lacking theoretical ground, and questionable if this is the minimal form: largely outstanding

**Reviewer Scores:**

Reviewer FXnf: 2
Reviewer g3Rd: 2
Reviewer WM28: 4
Reviewer e5yM: 4

I don't expect any of the scores to change if the discussion were to continue.

---

### Decision · Program_Chairs · 2026-01-26

Reject